# Grounding Generative Evaluations of Language Models in Unsupervised Document Corpora

Michael Majurski[1*]    Cynthia Matuszek[1]
[1]University of Maryland Baltimore County
michael.majurski@umbc.edu
cmat@umbc.edu

Reviewed on OpenReview: https://openreview.net/forum?id=EvtPh3Msol&noteId=EvtPh3Msol

## Abstract

Language Models (LMs) continue to advance, improving response quality and coherence. Given Internet-scale training datasets, LMs have likely encountered much of what users may ask them to generate in some form during their training. A plethora of evaluation benchmarks have been constructed to assess model quality, response appropriateness, and reasoning capabilities. However, the human effort required for benchmark construction is rapidly being outpaced by the size and scope of the models under evaluation. Having humans build a benchmark for every possible domain of interest is impractical. Therefore, we propose a methodology for automating the construction of fact-based synthetic data model evaluations grounded in document populations. This work leverages the same LMs to evaluate domain-specific knowledge automatically, using only grounding documents (e.g., a textbook) as input. This generative benchmarking approach corresponds well with human curated questions producing an ensemble Spearman ranking correlation of 0.91 and a benchmark evaluation Pearson accuracy correlation of 0.74 (model specific 0.82). This novel approach supports generating both multiple choice and open-ended synthetic data questions to gain diagnostic insight of LM capability. We apply this methodology to evaluate model performance on three recent documents (two post LM knowledge cutoff), discovering a surprisingly strong performance from Gemma-3 models on open-ended questions. Code is available at https://github.com/mmajurski/grounded-synth-lm-benchmark

## 1 Introduction

Recent advances in language models (LMs) have demonstrated significant capability improvements in diverse natural language processing tasks. This expansion in capability has primarily been enabled by training on Internet-scale corpora. While LMs generate fluent, plausible text, the outputs are not always grounded in the facts of a situation. As AI tooling improves professional workflows, rigorous evaluation is imperative to understand the domain-specific model capability.

Traditional multiple choice question (MCQ) benchmarks such as MMLU (Hendrycks et al., 2020) offer valuable general insights into model performance, but are static, public, and limited in scope. The fact that these benchmarks are static and public—and therefore can appear in training data—is a significant

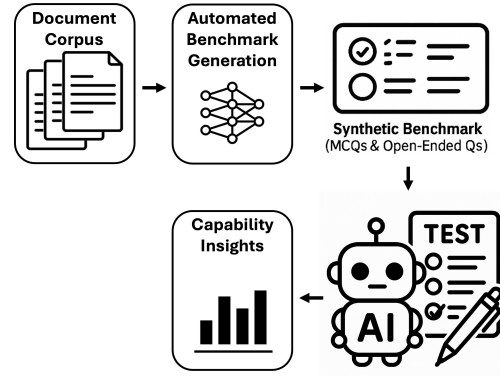

Figure 1: Generative evaluation pipeline: (1) users curate grounding documents, (2) LMs generate domain-specific questions, (3) model responses are evaluated.

weakness. Combined with the human effort required to

construct new high quality tests (Phan et al., 2025), it is unlikely evaluation construction can keep pace with LM development. This scaling challenge is especially acute for domain-specific professional applications, where the general benchmarks provide limited insight. Unfortunately, creating expert-validated benchmarks for every domain is impractical. New automated methods must generate accurate, comprehensive, diagnostically relevant evaluations grounded in authoritative corpora with minimal human intervention beyond document selection.

We propose an novel automated methodology for generating synthetic benchmarks grounded in authoritative documents. This approach reduces human effort and enables rapid adaptation to new domains by generating fact-based multiple choice or open-ended questions directly from appropriate grounding documents. This work is organized around three core questions:

1. How can LMs extract relevant information and generate high-quality questions grounded in domain documents?
2. Do these generative benchmarks replicate existing human-curated evaluations?
3. What limitations require mitigation for successful synthetic data evaluation of LM capability?

## 2 Related works

Our approach is motivated by task asymmetry where given grounding documents, an LM should reliably generate diagnostically relevant questions, while it is substantially harder to zero-shot answer those same questions. This information asymmetry motivates using LMs to generate benchmarks grounded in curated documents rather than relying only on model weight knowledge. Language models are routinely adopted as judges: extracting answer choices (Samuylova, 2024; Roucher, 2024), grading open ended responses (Ip, 2024; Oh et al., 2024), and justifying correctness (Cook et al., 2024). The most similar contemporary convergent work is YourBench (Shashidhar et al., 2025) synthetic replication of MMLU from grounding documents and AutoBencher (Li et al., 2025b) reinforcement learning optimization to discover gaps in LM knowledge. Leveraging grounding documents greatly improves synthetic data utility, as the methodology no longer relies on internalized LM knowledge.

Modern long-context LMs can process entire documents. Benchmarks like SecQA (Liu, 2023) and Pub-MedQA (Jin et al., 2019) leverage domain-specific textbooks and web pages. Full-document grounding generally outperforms summary-based systems (Bhat et al., 2023). Continued LM improvement, prompting strategies (Yadav et al., 2024), and topic breadth (Li et al., 2024) has further improved synthetic data. These methods also enhance downstream applications like retrieval-augmented generation (Balaguer et al., 2024).

LMs can serve as acceptable feature extractors when guided by detailed prompts. Studies show strong correlation between LM-based and human judgments of text quality (Chiang & yi Lee, 2023; Han et al., 2024). This aligns with the LM-as-a-Judge paradigm used for open-ended benchmark evaluation (Xu et al., 2023; Zheng et al., 2023). Beyond open ended answer correctness, LMs can extract other textual features. However, limitations in LM ground truthing remain, especially in tasks like information retrieval where relevance judgments are more complex (Soboroff, 2025; Szymanski et al., 2025). LM evaluation benchmarks generally use either multiple choice (MC) or open-ended (OE) formats. Open-ended questions require synthesis (Balepur et al., 2025) but depend on a judge LM for scoring (Xu et al., 2023; Myrzakhan et al., 2024), while MCQs are easier to grade but may inflate performance due to answer visibility (Robinson et al., 2022; Balepur et al., 2024). Frameworks like Inspect-AI (UK AI Security Institute, 2024) support both types with prompt templates and answer shuffling. Both formats are sensitive to rewording, leading to inconsistent outputs (Wang et al., 2024a), and newer frameworks aim to improve open-ended scoring precision (Bernard et al., 2024; Kamalloo et al., 2023).

## 3 Methods

Constructing synthetic data benchmarks from relevant domain-specific documents requires chunking into contexts, topic extraction per context, question and answer generation, and finally an evaluation framework to determine model capability on the newly minted benchmark (as shown in Figure 2). All LM interactions use default generation parameters, except we set temperature = 0.7 (o4-mini did not support setting temperature).

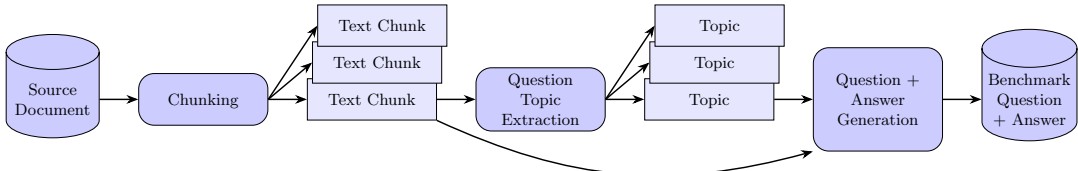

Figure 2: Generative benchmarking pipeline when used in production. During evaluation of the approach significantly more LM-Judge operations are included to evaluate question and answer quality.

**Document chunking into contexts**   While modern LMs have sufficient context windows to support whole document ingestion, chunking encourages the generated questions to be spread throughout the grounding document. An input document is first converted into markdown for interoperability with LMs. We use Docling (Auer et al., 2024) for conversion from pdf to markdown, and pandoc (MacFarlane) for conversion from html to markdown. Once in markdown, each document section is used as a context chunk for question generation. This chunking method is simple and relies on the document author crafting relevant sections. Enhancements to chunking and document data preparation will likely produce overall system improvements, but were not required to demonstrate baseline performance in this study. For instance, our approach only leverages the textual data; formatting and including richer sources of information from table, charts, and images may improve evaluation quality.

**Topic extraction**   For each extracted document context chunk, an LM is used to extract and propose any topics present which make high quality difficult questions for evaluating domain expert knowledge about the information in context chunks. This topic extraction leverages the same LM as the question generation process. We explored using an embedding model to cluster topics and de-duplicate, but empirically found no satisfying improvements, as the topic generation by LM does a reasonably good job of de-duplication. From manual observations, the only duplicate topics stem from different source document context chunks. This duplication is acceptable as the question generation is grounded in the context chunk; any duplicate topics across context chunks will produce different questions.

**Question generation**   Given (a) the document context chunk and (b) a single topic for a question, the generating LM is prompted to construct a high quality and difficult evaluation question to probe the knowledge and understanding of an expert in the domain. During question generation, the LM is required to write the question, correct answer, and a justification/explanation for why the correct answer is in fact right. The answer justifications are not used in the evaluation benchmark and simply exist to guard against generating incorrect answers. In summary, the single grounding document is broken into chunks of context, per-context relevant topics are extracted, then per-context/topic pair, and a single evaluation question is created. This results in a list of questions with correct answers grounded in (and derived from) the source document. This example drawn from Squadv2 (Rajpurkar et al., 2018) shows a question generated by Llama-3.3-70B-Instruct grounded in the context:

```
1    {"context": "In 1756 and 1757 the French captured forts Oswego and William Henry from the British
         . The latter victory was marred when France's native allies broke the terms of capitulation
          and attacked the retreating British column, which was under French guard, slaughtering and
          scalping soldiers and taking captive many men, women and children while the French refused to
           protect their captives. French naval deployments in 1757 also successfully defended the key
          Fortress of Louisbourg on Cape Breton Island, securing the seaward approaches to Quebec.",
2
3        "question": "In what year did French naval deployments successfully defend the key Fortress of
             Louisbourg on Cape Breton Island, securing the seaward approaches to Quebec?",
4
5        "choices":{"A": "1755",
6         "B": "1756",
7         "C": "1757",
8         "D": "1758"},
9
10       "answer": "C"}
```

**Multiple choice vs open-ended**   In this study, we explored both multiple choice and open-ended question generation, each requiring slightly differing LM prompts. While simply using the correct multiple choice answer option in an open-ended configuration works, we observed that open-ended questions enable more variety from

the generating model if prompts are modified. One study goal was to determine if any differences manifested between multiple choice and open-ended questions for synthetic data benchmarking. The difference in response grading requirements between open and multiple-choice favors MCQs for fast and simple exploration of model weaknesses.

## 3.1 Benchmark evaluation

Given a list of questions and answers, an LM evaluation framework (Inspect-AI (UK AI Security Institute, 2024)) is used to determine performance for the models under evaluation. The benchmark consists of many questions with associated answers. Open ended questions are graded using the default Inspect-AI framework support which uses a narrowly scoped LM-Judge to compare a candidate model output to the ground truth answer to determine correctness. The resulting benchmark evaluation accuracy is the average number of correct answers divided by the total number of questions. LMs are evaluated against both the synthetic evaluation benchmark and the original extractive QA human written questions lightly rewritten to be answerable without the context. The overall accuracy per evaluated model can be compared for both the synthetic and original questions across multiple datasets of grounding documents. The text of the various benchmark questions and answers is never directly compared, only the resulting LM performance. Figure 3 showcases the performance of synthetic multiple choice questions averaged across datasets and generating models, with a Spearman ranking correlation of 0.907, demonstrating slight model over-performance on the synthetic benchmark.

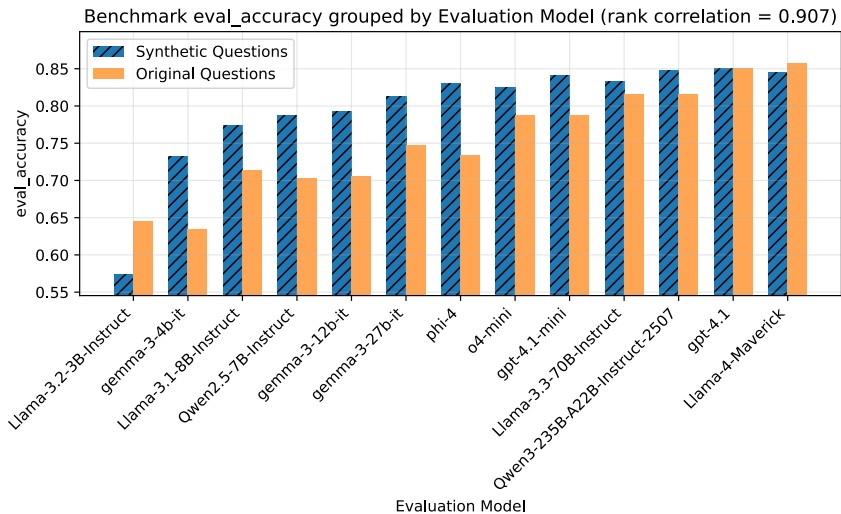

Figure 3: Comparison of average LM performance across NLP datasets with the multiple choice question synthetic data benchmarks created by the ensemble of bolded model in Table 2.

## 4 Evaluation methodology

Evaluating the quality and utility of the newly minted synthetic data benchmark built upon grounding documents leveraged Natural Language Processing (NLP) datasets where each entry contains a context paragraph, human written question, and correct answer. This provides a directly comparable result, both for the overall benchmark accuracy (between human and grounded synthetic data) and for the rank ordering of those models that the benchmark is evaluated on. For each dataset a variety of LMs are used to generate the synthetic benchmark. Per-question quality features and benchmark-level question diversity is extracted to compare against the human written questions. Finally, an automatic quality control pass is completed to grade both question and answer correctness/validity, in order to investigate and characterize which benchmark-generating models can write valid question and answer pairs.

## 4.1 Datasets

Evaluating the validity of generated synthetic questions requires source documents which have human annotated question/answer pairs to compare against. The following NLP datasets were evaluated: (a) Squadv2 (Rajpurkar et al., 2018) (b) HotpotQA (Yang et al., 2018) (c) TrivaQA-web (Joshi et al., 2017) (d) NaturalQuestionsShort (Kwiatkowski et al., 2019) (e) PubMedQA (Jin et al., 2019) (f) BoolQ (Clark et al., 2019) (g) FermiQA (Kalyan et al., 2021) (h) MS-MARCO-QA (Bajaj et al., 2016) (i) MusiqueQA (Trivedi et al., 2022) (j) NarrativeQA (Kočiskỳ et al., 2018) (k) 2WikiMultiHopQA (Ho et al., 2020), and (l) SecQA (Liu, 2023).

Table 1: Example human question reformatting

| **Context:** Super Bowl 50 was an American football game to determine the champion of the National Football League (NFL) for the 2015 season. The American Football Conference (AFC) champion Denver Broncos defeated the National Football Conference (NFC) champion Carolina Panthers 24–10 to earn their third Super Bowl title. The game was played on February 7, 2016, at Levi's Stadium in Santa Clara, California. As this was the 50th Super Bowl, the league emphasized the "golden anniversary" and temporarily suspended Roman numerals in the logo. |
| --- |

| Original Question | Original Answer | Reformated Question | Answer Choices |
| --- | --- | --- | --- |
| What day was the game played on? | February 7, 2016 | On which specific date did Super Bowl 50 take place at Levi's Stadium in Santa Clara, California? | (A) January 7, 2016
**(B) February 7, 2016**
(C) February 14, 2016
(D) January 14, 2016 |
| If Roman numerals were used, what would Super Bowl 50 have been called? | Super Bowl L | What would the 50th Super Bowl have been called under the traditional Roman-numeral naming? | (A) Super Bowl XL
(B) Super Bowl XLVIII
(C) Super Bowl XLIX
**(D) Super Bowl L** |

These datasets collectively contain tens of thousands of context, question, answer tuples. A random subset of 100 samples per dataset was selected as the grounding context document. For certain datasets, preprocessing was required to clarify, curate, and reformat the source human written questions into self-contained questions. For some questions this required disambiguation so the question is answerable without the context. For the multiple choice evaluation, three believable incorrect answer options needed to be created. The PubMedQA (Jin et al., 2019) and BoolQ (Clark et al., 2019) datasets required conversion from yes/no questions into multiple-choice or open-ended. Only the SecQA (Liu, 2023) dataset already contained all of the required information and was not pre-processed. All other datasets went through an LM based reformatting operation correct these deficiencies. Table 1 provides an example of the reformatting operation for a few questions drawn from a single context of the Squadv2 dataset. This disambiguation rewrite was validated using both an LM-Judge and embedding cosine similarity to ensure the questions were not unduly altered. Throughout the rest of this work, original questions refers to these lightly edited and disambiguated version of the human authored questions.

Three different models (Llama-4-Maverick, gemma-3-27b-it, o4-mini) were evaluated for this reformatting operation. The remainder of this paper shows the Llama-4-Maverick reformatting (see Table 4 in Appendix F for the minimally different results of other reformatting models).

## 4.2 Question generation models

To explore the impact of model capability on successful generation of synthetic data benchmarks we tested a variety of language models during the synthetic benchmark generation phase (see Table 2). All open-weight models are available through HuggingFace (hug, 2024) and the closed-weight models were accessed through paid APIs. Additional results are reported for an ensemble of the bolded

Table 2: Question generating language models

| | |
| --- | --- |
| Google | gemma-3-4b-it (Team et al., 2025)
gemma-3-12b-it (Team et al., 2025)
**gemma-3-27b-it** (Team et al., 2025) |
| Meta | Llama-3.2-3B-Instruct (met, 2024)
Llama-3.1-8B-Instruct (Grattafiori et al., 2024)
**Llama-3.3-70B-Instruct** (Grattafiori et al., 2024)
**Llama-4-Maverick-17B-128E-Instruct** (lla) |
| OpenAI | **o4-mini** (ope, 2025)
**gpt-4.1** (ope, 2025)
gpt-4.1-mini (ope, 2025) |
| Microsoft | phi-4 (Abdin et al., 2024) |
| Qwen | Qwen2.5-7B-Instruct (Yang et al., 2024)
**Qwen3-235B-A22B-Instruct-2507** (Team, 2025) |

models where all generated questions from model ensemble were mixed together as an evaluation benchmark. During benchmark creation, the same model is used both for topic extraction and question generation.

### 4.3 Evaluating against human questions

The reformatted human annotated question/answer pairs are used to compare the grounded synthetic data benchmark performance against the human. Ideally, the synthetic benchmark from grounding documents would produce the same LM capability measurements as the human-curated benchmark for any given model under evaluation. As such, to validate our approach, various LMs of differing sizes are evaluated on both the human and synthetic benchmarks. Comparison of pure benchmark evaluation accuracy is performed using the Pearson accuracy correlation between human and synthetic benchmark accuracy numbers. Evaluating the comparative rank ordering of models uses the Spearman rank correlation of relative model performance on the given benchmark dataset. These two measures compare both absolute performance alignment and relative rank order preservation/stability.

### 4.4 Generated question quality characterization

Per-question quality features and benchmark dataset-level features are computed to determine the overall quality of the synthetic benchmark. Owing to the low level of human oversight and annotation time for creating the synthetic data benchmarking, LMs extract these somewhat imprecise features. This relies on the idea that with enough guidance LMs can be good alternatives to human evaluation (Chiang & yi Lee, 2023; Han et al., 2024) and borrows from standard practice LLM-as-a-Judge grading of open-ended benchmark responses (Xu et al., 2023; Zheng et al., 2023). Llama-4-Maverick extracts the following information about each generated question:

1. **question token count**: number of tokens in the question
2. **question clarity**: how clear and comprehensible is the question in isolation without the associated context
3. **question groundedness**: how grounded is the question in the context
4. **answer correctness**: how correct is the answer given the context
5. **answer explanation validity**: how valid and logically consistent the answer explanation is

In addition to per-question measurements, question diversity is measured over each benchmark as the average cosine similarity between all questions (using sentence-transformers/all-MiniLM-L6-v2 as the embedding model). The `answer_correctness` and `answer_explanation_validity` numbers flag which generating models are incapable of producing diagnostically useful questions.

## 5 Results

All experiments and development were conducted on a single 8×H100 on-premise machine. Development consumed approximately twice the compute of final evaluations. Final compute was split between synthetic dataset generation and benchmark evaluation via Inspect-AI. The most expensive model, gpt-4.1 ingests about 3.6M input and 1.1M output tokens for an average evaluation run ( $16), with compute evenly divided between generative dataset creation and evaluation.

### 5.1 Answer correctness

Synthetic benchmark answers must be accurate, a task made easier via the grounding documents that supply the necessary information. To verify generated answer accuracy, an LM scored each question-answer pair and its justification (1-10 scale) using the full context as described in Section 4.4. As shown in Figure 4, reasonably sized models scored above 9.5 in accuracy, with smaller models performing noticeably worse. While model size/capability generally correlates with desirable properties for synthetic data generation, notable exceptions (like gpt-4.1-mini) are discussed in Figure 14 of Appendix F.

### 5.2 Question clarity and groundedness

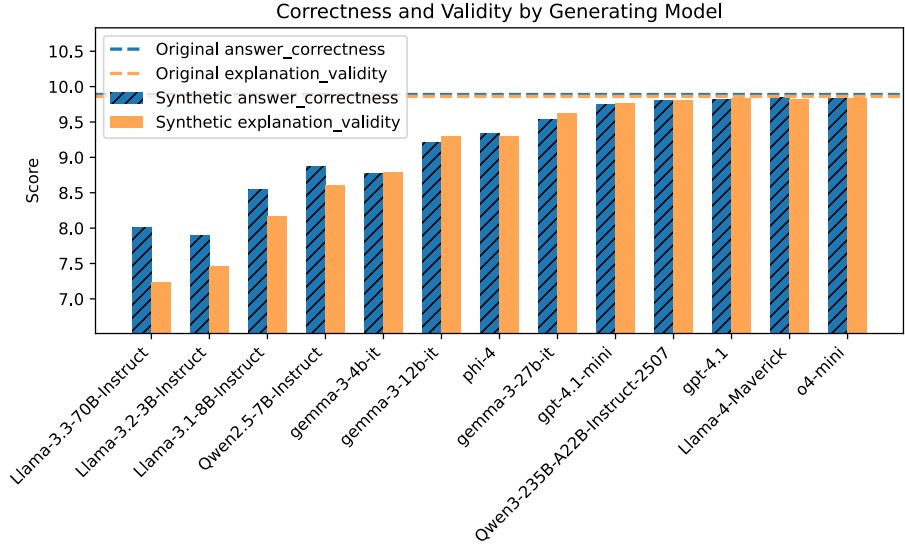

Figure 4: Synthetic data generated question correctness and explanation validity on a scale of 1 to 10.

Question clarity is inherently tied to difficulty, as simple questions have the potential to be clearer, easier to understand, and easier to answer. Conversely, a needlessly opaque question is harder to answer while providing minimal additional diagnostic relevance, though a model's ability to reason through more opaque questions indicates general capability. This study found that most generated questions have acceptable clarity without significant variation between the generating models. Most generated synthetic questions are longer and more complicated than the original questions. This increased average question complexity combined with similar levels of question clarity suggests most synthetic questions are well written. Manual random spot inspection supports this as we observed few failures.

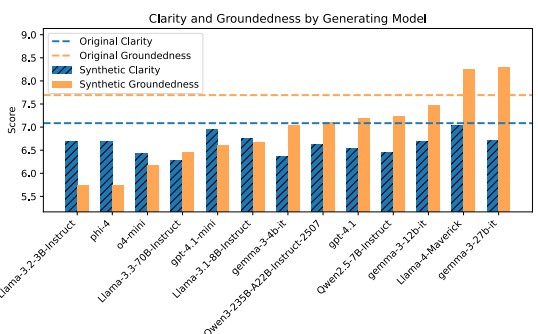

Figure 5: Generated question clarity and groundedness compared against the original questions.

How grounded in the source material (context chunk) a given question is indicates how much outside information both the generation and evaluation models must leverage to correctly interpret what is being asked and answer. Low grounding indicates the question/answer pair cannot be found within the context chunk, or that significant external information is required. High grounding means all necessary information is found within the context chunk. Therefore the generated question is testing for information aligned with and present in the context. Figure 5 highlights the question clarity and groundedness scores for the question generating models. While most LMs generate similar clarity values, the trends in groundedness score are more complex. Some capable models (like Llama-4-Maverick) score well, other capable models (like o4-mini) score worse, with smaller models in between (gemma-3-4b-it). We observed two grounding failure modes: either the model hallucinates irrelevant content absent from the context chunk (phi-4 commonly did this), or the generating model uses knowledge in its weights to expand upon the information presented in the context (o4-mini). See Appendix G for an example from phi-4. Models of medium size perform well on groundedness, simply reformatting information presented in the context.

## 5.3 Impact of question length

The synthetic data questions in this study tend to be longer than the original human-written questions. The generating model is prompted to include all information required to disambiguate what the synthetic question is asking about, causing the questions to be longer and more detailed than the original dataset.

Figure 6a highlights this trend. Some generating LMs take this to an extreme, causing the synthetic questions to become easier for LMs to answer correctly. We hypothesize this happens because the additional detail in the question places the LM under evaluation in the right "state of mind" so to speak, in the representation vector space. This effect is seen in Figure 6b where benchmark evaluation accuracy over-performance (synthetic $-$ original accuracy) is plotted. There is a rough trend where longer questions over-perform the $y = 0$ expected trend line. It is worth noting that this effect does not show up in answer length (see Appendix C for detail and examples).

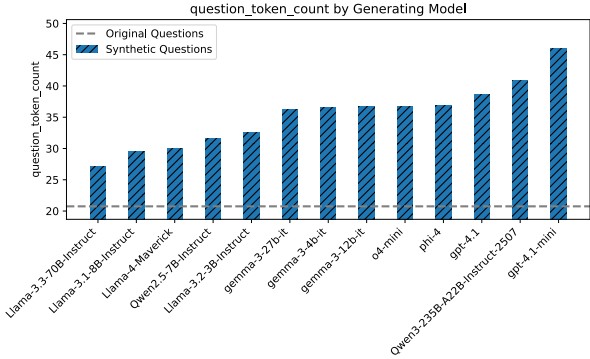

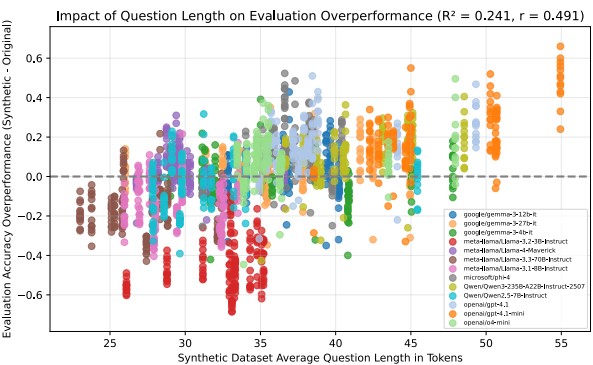

(a) Question token count by generating model. Most models are more verbose than humans, but certain models take this to an extreme, producing very long questions.

(b) Scatterplot documenting evaluation accuracy over-performance on long questions. Models like gpt-4.1-mini plot well above the trend line.

Figure 6: Impact of question length on evaluation accuracy over-performance.

## 5.4 Benchmark question diversity

During synthetic question generation, the explicit topic extraction step provides a backstop against degenerate questions which only focus on the most obvious elements of a context chunk. To compare the variety and coverage of the generated questions we compute the average cosine similarity between all question pairs (excluding self pair) over the generated synthetic dataset. While most generating models produce similar diversity scores closely aligned with the human questions, some highly capable models (such as gpt-4.1) show comparatively worse question diversity while performing well in clarity, correctness, and groundedness. Figure 10 in Appendix D shows the question diversity per generating model along with the original question diversity.

## 5.5 Benchmark model accuracy correlation

Ideally, we want synthetic benchmarks to correlate well with the evaluation accuracy measured by human curated questions. This ends up being true for some generating models, but not all. Figure 7 shows the evaluation accuracy scatterplot comparing the reformatted human questions against the synthetic generated benchmark questions for the five bolded ensemble models in Table 2. This ensemble produces

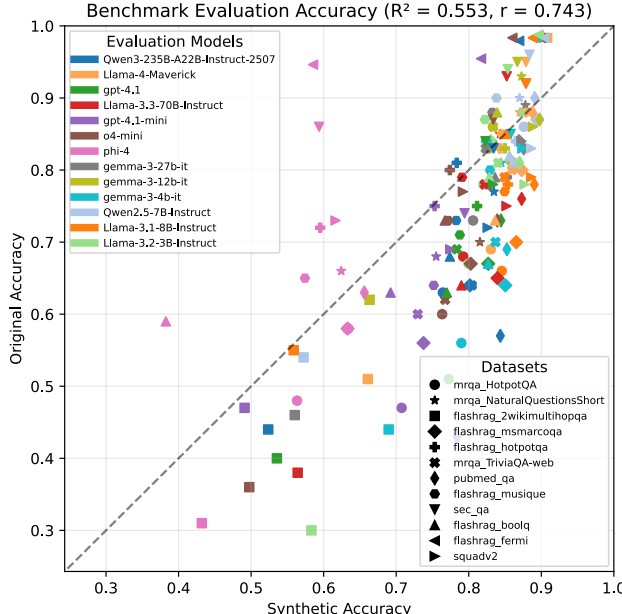

Figure 7: Benchmark evaluation accuracy correlation generated by model ensemble demonstrating reasonable correlation between human and synthetic evaluation. Pearson accuracy correlation 0.743.

a Pearson accuracy correlation of 0.743 between the evaluation accuracies for synthetic and human benchmarks (Figure 15b). Each plot marker is the average evaluation accuracy for a given model-dataset combination. While this model ensemble produces reasonable overall accuracy correlation, occasionally specific LMs generate undesirable benchmarks. Figure 14 in Appendix F showcases a failure where the synthetic benchmark generated by gpt-4.1-mini contains overly easy questions where all models perform near 100% evaluation accuracy (Pearson accuracy correlation 0.276).

### 5.6 Benchmark model rank correlation

While direct Pearson correlation between human and synthetic benchmark evaluation accuracy is desirable, evaluations are often used to compare models. Therefore, a more important metric is the model rank order stability measured by Spearman rank correlation. Figure 3 plots the evaluation model performance across all datasets, comparing the synthetic and human accuracy Spearman ranking correlation of the evaluation model capability. Pearson accuracy correlation translates into Spearman ranking correlation and relative model order stability. Similarly, the overly easy gpt-4.1-mini synthetic benchmark in Figure 14 produces an odd rank ordering (Spearman correlation 0.753) in Figure 15b with a very flat accuracy curve. While ensembling various question generating LMs together will partially mitigate this effect, it is worth validating the set of generating models against the datasets in this paper to confirm the ensemble is not negatively impacting overall synthetic benchmark quality. The tendency of certain generating models to produce overly long, detailed questions that LMs find easier to answer may be the cause of the observed benchmark over-performance.

### 5.7 Multiple-choice vs open-ended questions

During this study, both multiple choice and open-ended questions were evaluated. None of our findings change whether using multiple choice questions for simplified evaluation benchmark grading, or open-ended questions to produce harder evaluation benchmarks where knowledge synthesis is required. The models that are good at creating synthetic multiple choice questions are also good at creating open ended questions. The models which create overly easy questions by including too much detail (like phi-4) do so for both open ended and MCQ, with the long question problem becoming even more pronounced for open ended questions as shown in Figure 11 in Appendix E.

Table 5 presents Pearson benchmark evaluation accuracy correlation and Spearman rank order correlation between the human and synthetic evaluations. Llama-4-Maverick was unexpectedly dominant for multiple-choice in these final Table 5 results, even outperforming the ensemble of models. Similarly gemma-3-23b-it over-performed the ensemble for open-ended questions.

Table 3: Pearson accuracy and Spearman rank correlation between synthetic and human benchmarks

| Metric | Pearson Accuracy Correlation | | Spearman Rank Correlation | |
|---|---|---|---|---|
| **Generating Model** | **Multiple Choice** | **Open Ended** | **Multiple Choice** | **Open Ended** |
| Ensemble | 0.7434 | 0.7238 | 0.9066 | 0.6099 |
| Llama-3.2-3B-Instruct | 0.5444 | 0.7288 | 0.2527 | 0.4176 |
| **Llama-3.3-70B-Instruct** | 0.6807 | 0.6992 | 0.8626 | 0.4725 |
| Llama-3.1-8B-Instruct | 0.5484 | 0.7402 | 0.8901 | 0.4780 |
| **Llama-4-Maverick** | **0.8232** | 0.7833 | **0.9231** | 0.6758 |
| gemma-3-12b-it | 0.6879 | 0.6521 | 0.7363 | 0.5000 |
| **gemma-3-27b-it** | 0.7325 | **0.8192** | 0.8846 | **0.8516** |
| gemma-3-4b-it | 0.6495 | 0.7771 | 0.5549 | 0.7308 |
| **gpt-4.1** | 0.5784 | 0.5071 | 0.7967 | 0.4341 |
| gpt-4.1-mini | 0.2759 | 0.2505 | 0.7527 | 0.4560 |
| **o4-mini** | 0.6687 | 0.5290 | 0.8187 | 0.5495 |
| phi-4 | 0.3489 | 0.3936 | 0.5989 | 0.4725 |
| Qwen2.5-7B-Instruct | 0.5901 | 0.6018 | 0.4286 | 0.6099 |
| **Qwen3-235B-A22B-Instruct-2507** | 0.6084 | 0.6017 | 0.7857 | 0.5495 |

### 5.8 Case study on academic papers

To demonstrate our approach in a more realistic setting, we selected three documents (two of which are post-LM training knowledge cutoff): "Current Solutions and Future Trends for Robotic Prosthetic Hands" (Mendez et al., 2021), "Recent Advances in Large Language Model Benchmarks against Data Contamination: From Static to Dynamic Evaluation" (Chen et al., 2025), and the recently released "America's AI Action Plan" (AiP). The documents were chunked into roughly 30 chunks each. "Current Solutions" averaged 7.7 topics per context for a total of 210 questions. "Recent Advances" averaged 6.9 topics per context for a total of 194 questions. "AI Action Plan" produced a total of 235 questions. For MCQ, the synthetic benchmarks' Spearman rank correlation is 0.911, for OE the rank correlation is 0.969. These correlations reflect the trend of overall model capability, as both source documents are from post training cutoff, so models forced to puzzle out what the likely answers are. Unlike earlier results where the correlation was a comparison to the human curated questions, these correlations only compare the LM benchmark performance between the two academic papers. Under MCQ the gemma-3 models perform middle of the pack on these two benchmarks, but under OE questions the 12b and 27b outperform all other models. See Figure 16 in Appendix H for full rank ordering and examples of the generated questions.

### 5.9 Replicating a subset of MMLU-Pro

The datasets used herein to validate the generative benchmarking approach were never designed to evaluate general LM knowledge. To alleviate this weakness and demonstrate the capability of this generative benchmark creation approach, we replicated a subset of the MMLU-Pro benchmark (Wang et al., 2024b) using creative commons textbooks from OpenStax (Stafford & Flatley, 2018). Six of the categories from MMLU-Pro were replicated by using a related textbook as the grounding context: biology (Clark et al., 2018), chemistry (Flowers et al., 2019), computer science (Franchitti, 2024), math (Abramson, 2021), physics (Urone & Hinrichs, 2022), and psychology (Spielman et al., 2020). We used `gpt-oss-120b` to construct the generative benchmarks. Each textbook was converted into markdown, chunked into 4k token blocks, and ingested into the generative benchmarking pipeline. The resulting generative benchmarks were evaluated using the full suite of models in Table 2. Figure 8 demonstrates the correlation between the generative replication of each MMLU-pro subset and the original human-curated benchmark results.

The generative replication of MMLU-pro benchmark produces a Pearson accuracy correlation of 0.78 with

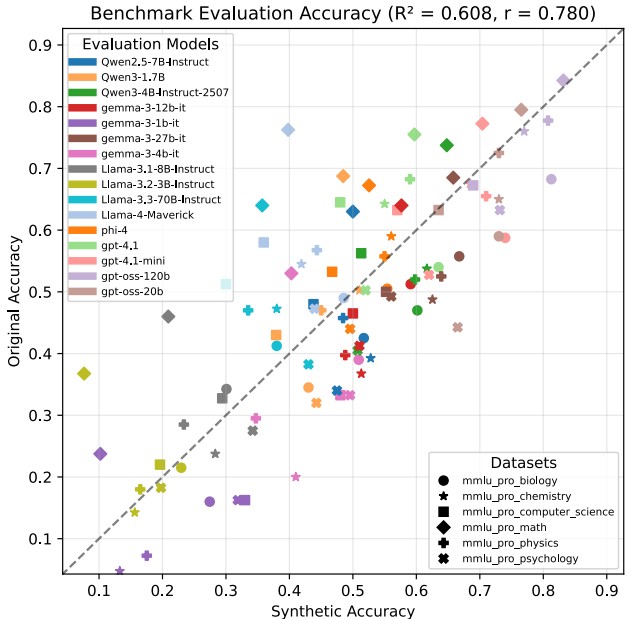

Figure 8: Benchmark evaluation accuracy correlation generated by model ensemble demonstrating reasonable correlation between human and synthetic evaluation. Pearson accuracy correlation 0.78.

the original benchmark (Spearman rank correlation of 0.83). Given that the generated questions were created using only relevant textbooks where question topics do not align, the resulting level of agreement between human curated and grounded generative benchmarking is surprising. Additionally, the accuracy aligns well with underlying model capability as measured by the human-curated MMLU-Pro benchmark.

To validate the quality of the generative questions, the authors manually reviewed 120 questions (20 each randomly selected from each MMLU-pro subset). This evaluation probed two elements, the question/answer pair correctness and grounding in the source document context. We found that 111 of 120 questions had completely accurate generative question/answer pairs, meaning the question was well formed and the answer was correct. This yields a noise floor for generative benchmarking approach of approximately 7.5%. 104 of

120 questions had fully supported grounding in the document. A strong majority of the grounding failures fell into two categories: 1) the question writing model adding relevant correct information that was missing from the document, or 2) the generative question referencing the document and not being fully independent.

An example high quality generative question from the physics subset is as shown:

> **Generative Question:** Why can a pion exchanged between a proton and a neutron transmit the strong nuclear force without being directly observable, and how does its fleeting existence satisfy $\Delta E \cdot \Delta t \geq \hbar/2$ while leaving the total energy–momentum of the nucleon system unchanged?
>
> **Generative Answer:** Because the pion exists only for a time $\Delta t \leq \hbar/(2\Delta E)$ (with $\Delta E = mc^2$), allowing a temporary energy borrowing that is repaid when the pion is reabsorbed, so the nucleon system's total energy–momentum stays conserved, rendering the pion virtual and unobservable.
>
> **Human Evaluation:** answer is Correct and Supported by the grounding context.

We also provide an example of a question where the system failed to produce a correct answer:

> **Generative Question:** Based on the descriptions provided, what are the chemical compositions of the protective passivation films that develop on the Statue of Liberty's copper surface and on stainless steel, how does the text characterize their adherence to the underlying metal, and does it mention any self-healing capability under cyclic wet-dry exposure?
>
> **Generative Answer:** The copper patina consists of copper oxides transformed by $SO_3$, $CO_2$, and $H_2O$ into a blue-green protective film, stainless steel's passivation layer is chromium oxide, both are described as adherent protective layers, and the text does not address self-healing under wet-dry conditions.
>
> **Human Evaluation:** answer is Incorrect and Unsupported because the question relies on information present in the context. Additionally, the answer formulation changed the context text from "The chromium tends to collect near the surface, where it corrodes and forms a passivating **an oxide layer** that protects the iron" to the wrong answer of "chromium oxide".

A more detailed analysis of the human-evaluated question set is provided in Appendix B. The MMLU-pro replication used `gpt-oss-120b` for question generation. It is possible that using a stronger frontier model would reduce the generation and grounding error rate observed.

### 5.10 Limitations

This work relies on a simple and possibly non-optimal document chunking approach. Extensions which cluster related topics or provide more fine-grained control over the information present in each chunk could improve the granularity and specificity of the generated benchmark. Including information found in tables, charts, and images would provide richer information sources that benchmark creation may take advantage of. Additionally, improving the nuance and difficulty of the questions significantly improves LM capability measurement. This likely requires careful management of the context information provided to the generating LM. This approach is not yet suitable for evaluating the absolute frontier of model capability, but many applications rely on weaker models with lower inference costs. Finally, except for the subset of MMLU-Pro replicated using Creative Commons textbooks, the datasets used in this study to validate correlation with human written questions are all NLP extractive question answering type datasets. This limits the quality of the synthetic benchmarks as the grounding documents are fairly simplistic, especially compared to the two academic papers used in Section 5.8. Only the `PubMedQA` and `sec_qa` datasets come close to the topic diversity and complexity of the example academic papers. All other validation datasets rely upon more general factual knowledge.

## 6    Conclusion

We introduce a methodology for creating synthetic benchmarks to evaluate language model (LM) capabilities grounded in high-quality document corpora selected by practitioners. This ensures the generated benchmarks are directly relevant to their specific domains and interests. This approach extends general-purpose human curated LM benchmarking into specialized domains, allowing for the identification of strengths and weaknesses that may not be apparent in broader evaluations.

The majority of the invalid question/answer pairs stemming from either missing information or grounding failures could be rectified by an adversarial verifier running within the question generation loop. In future work we plan to explore this research direction both for generative benchmarking and dynamic context grounding.

Our method supports automatic generation of both multiple-choice questions (optimizing for ease of grading) and open-ended questions (optimizing for diagnostic relevance), allowing users to choose based on their evaluation goals. Validation against human-curated NLP dataset achieves an ensemble Spearman rank correlation of 0.91 and a benchmark Pearson accuracy correlation of 0.74 between synthetic and human curated questions, suggesting that our approach is consistent with human-generated question/answer benchmarks. Additionally, we identify a potential synthetic data generation failure mode: very long questions can artificially inflate model performance, likely by overspecifying information related to the answer. This methodology enables the creation of targeted, relevant benchmark evaluations facilitating deeper exploration into the domain specific capabilities and limitations of Language Models deployed for complex knowledge work.

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

# A    Expanded related works

This approach draws inspiration from *NP-completeness*, where easy to verify solutions may be hard to generate. The LM under evaluation has a harder job than the LM that converts existing documents into appropriately formatted test questions with accompanying answers. The evaluation LM must produce the correct information from just the question; by contrast, the benchmark creation LM has the source material. In other words, given the grounding paragraph, a language model should be much more capable at converting that textual information into a multiple choice or short answer question than it would be at zero-shot answering the very same question. This task difficulty discrepancy provides hope that this approach will produce viable results long term. Asking an LM to generate diagnostically useful evaluation benchmarks drawn only from information stored in model weights is difficult, but conversion of information from a high quality source into test questions is attainable.

This work draws on the rich history of language model evaluation, where LMs are already being used to judge and score the results of human curated benchmarks. When scoring outputs of an LM producing a multiple choice answer, it is often useful to have another LM reason about and extract the specified `[A,B,C,D]` value, instead of hoping the model under evaluation only produced the exact match answer letter (Samuylova, 2024; Roucher, 2024; Wolfe, 2024). This process becomes even more important when evaluating short answers from LMs instead of multiple choice responses, or when scoring the responses and providing reasoning as to why an answer was correct or incorrect (Ip, 2024; Oh et al., 2024; Cook et al., 2024). This idea of automating question answer generation has been approached before. The most similar contemporary work is YourBench by Shashidhar et al. (Shashidhar et al., 2025), which leverages the convergent idea of grounding documents supporting LM benchmark generation, demonstrating the synthetic replication of MMLU. Additionally, AutoBencher by Li et al. explores leveraging synthetic data (supported by privileged information) for model evaluation with RL to optimize the generated evaluation for discovering gaps in LM capability (Li et al., 2025b). The use of grounding documents or privileged information greatly improves synthetic data benchmark utility, as the evaluation methodology no longer relies on simply asking an LM for questions (and answers).

**Question from context and answer**    Early Question Answer Generation (QAG) approaches attempted to solve the question creation from a context + correct answer input. These approaches include improvements in selecting better LM samples (Yuan et al., 2022), increasing sample diversity over distributional shifts (Shinoda et al., 2020), explorations of interrogative prompting against the context (Eo et al., 2023), and evaluating round trip information consistency (Alberti et al., 2019). These early methods sometimes fine-tune generation models to improve task performance, though largely that is only done with smaller models (Ushio et al., 2023; Zhou, 2024). Some work focused on fine-tuning specialty models to improve option generation for multiple-choice questions (Zhou, 2024). Other early work during the shift to general purpose LMs involved leveraging those language models directly as data generation sources. This took the form of prompting the model to write questions (without grounding context) (Bai et al., 2023; Tan et al., 2024; Perez et al., 2023; Zhang & Wang, 2022), or configuring an LM-as-evaluator/interrogator dynamically interacting with the LM under evaluation to explore capability in a reference-free manner (Bai et al., 2023). These approaches attempt to address the dataset leakage problem where benchmarks, once published, become part of the training dataset for the next round of models. As the capability of LMs increases over time, setting up adversarial LM-evaluator vs LM-evaluatee systems is becoming more feasible (Li et al., 2025a). Nevertheless, extending those systems such that the LM-evaluator has grounding documents to pull from should strengthen the performance.

**Scaffolding question generation with grounding documents**    With the rise of general purpose token-tumblers like ChatGPT, longer and more complex grounding documents become increasingly useful. There is no longer a need to constrain synthetic benchmark creation to a short paragraph context + correct answer input; the LM can operate on larger and more complex document chunks, with less pre-processing required to make the input context data useful. Contemporaneous work YourBench (Shashidhar et al., 2025) and AutoBencher (Li et al., 2025b) can leverage lightly-filtered web-pages as the context, chunking the data into rough paragraphs for ingestion into the question generation and grounding system. The SecQA benchmark was created partly through synthetic data grounded in an open-source computer security

textbook (Liu, 2023). Similarly (though much earlier), the PubMedQA benchmark was created through human and synthetic data generation grounded in biomedical research paper abstracts (Jin et al., 2019). Some research demonstrates that operating purely on document summaries produces inferior generated questions compared to leveraging the full document (Bhat et al., 2023). With modern long-context LMs, summarization is becoming less of a problem, as models can manipulate significant volumes of text within their attention context. Synthetic data question generation systems have seen rapid improvement. This is partly driven by capability improvements in the LMs powering synthetic data generation. However, there has also been exploration into improvements in prompting for diverse question generation (Yadav et al., 2024) and breadth of topic generation for chatbot arena style questions (Li et al., 2024). Enhancing retrieval augmented generation (RAG) systems for domain specific applications has also used synthetic data question generation to compare baseline RAG to fine-tuning (Balaguer et al., 2024).

**LM as feature extractors**   There are many aspects of text that one might want to quantify, however imprecisely. However, devoting the required copious human annotation hours is just impractical. Current LMs end up being acceptable feature extractors, when provided detailed prompts. Chiang and Lee demonstrate that LMs can substitute for human evaluators in assessing text quality with strong correlation in the results (Chiang & yi Lee, 2023). Similarly, Han et al. leverage LM evaluators for judging quality and find that results correlate highly with human judgments (Han et al., 2024). The use of LMs to extract features from text is not to dissimilar from using the LM-as-a-Judge (Xu et al., 2023) pattern for large scale open ended capability evaluations (Xu et al., 2023; Zheng et al., 2023; Szymanski et al., 2025). The LM is capable of grading open ended evaluation benchmark questions for correctness; therefore determining features other than answer correctness from a text sample is viable. However, there are limits on what LM-as-a-Judge is capable of ground truthing, especially in fields like information retrieval (Soboroff, 2025; Szymanski et al., 2025) where LMs under-perform on more complex relevance judgments. Relying on LMs to extract properties like answer correctness given sufficient scaffolding and context is a well-used pattern.

**Multiple choice vs open ended**   When constructing knowledge or fact based evaluation benchmarks, there are two broad categories of answers: multiple choice and open ended. Open-ended questions provide the model less information and require synthesis (Balepur et al., 2025), but at the cost of needing another LM to evaluate the correctness of the results (LM-as-a-Judge (Xu et al., 2023)) (Myrzakhan et al., 2024). Multiple choice question (MCQ) results can be parsed and graded more simply. Modern evaluation frameworks like Inspect-AI (UK AI Security Institute, 2024) enable shuffling answer options and contain default prompts for both open-ended and MCQ evaluations. With MCQ, having the answers visible to the LM can artificially inflate performance (Robinson et al., 2022), even going as far as the model identifying the correct answers without seeing the question (Balepur et al., 2024). Both open-ended and multiple choice questions have weaknesses to question reformatting and phrasing where semantically identical questions produce different answers (Wang et al., 2024a). The simplest form of open-ended evaluation relies on a judge LM, though other frameworks have been proposed which attempt to improve the precision of open-ended evaluations (Bernard et al., 2024; Kamalloo et al., 2023).

## B   Replication of MMLU-pro

Because LM-as-a-judge was used throughout our experiments, a human evaluation of a subsampled set of questions generated to replicate MMLU-pro was performed in order to validate results. The authors took a random 20 questions from each evaluated subset of MMLU-pro (`biology, chemistry, computer science, math, physics,` and `psychology`) and manually graded the question/answer pair correctness and grounding against the source context from textbooks. Each question-answer pair was labeled with each of correct/incorrect and supported/unsupported by the context. Answer correctness was often negatively impacted by grounding failures, such that a question was missing information from the context required to be correctly answered. Per-domain details are as follows:

- **Physics**: 19/20 questions correct, with the incorrect `Question 11` incorrectly canceling the torque of a pivot. 14/20 questions were fully supported by the grounding context; of the remaining 6, the most common failure was the question-writing model relying on model parameter knowledge to fill in

common knowledge which was not explicitly included in the grounding context. See Appendix B.1 for a per-question breakdown of the grounding failures in the `physics` category.

- **Biology**: 20/20 question/answer pairs correct, and 20/20 with correct and valid grounding in the source contexts.

- **Chemistry**: 18/20 question/answer pairs correct, 15/20 questions were fully grounded in the context. Both incorrect answers failed due to the question-writing model not including all required information in the question and or modifying minor details from the context. For example:

> **Question 13:** Based on the descriptions provided, what are the chemical compositions of the protective passivation films that develop on the Statue of Liberty's copper surface and on stainless steel, how does the text characterize their adherence to the underlying metal, and does it mention any self-healing capability under cyclic wet-dry exposure?
>
> **Answer:** The copper patina consists of copper oxides transformed by $SO_3$, $CO_3$, and $H_2O$ into a blue-green protective film, stainless steel's passivation layer is chromium oxide, both are described as adherent protective layers, and the text does not address self-healing under wet-dry conditions.
>
> **Human Evaluation:** Incorrect and Unsupported because the question relies on information present in the context. Additionally, the answer formulation changed the context text from "The chromium tends to collect near the surface, where it corrodes and forms a passivating **an oxide layer** that protects the iron" to the wrong answer of "chromium oxide".

The five Chemistry grounding failures resulted from the question writing model added additional correct information from its parameter knowledge to flush out the question. For example:

> Question 7: Using $\Delta G = \Delta H - T\Delta S$, predict how the spontaneity of NaCl melting changes when the temperature is raised from 500 °C to 700 °C, and explain why the entropy contribution becomes more decisive at the higher temperature.
>
> Answer: Nonspontaneous at 500 °C ($\Delta G > 0$) and spontaneous at 700 °C ($\Delta G < 0$) because the larger $T\Delta S$ term at the higher temperature outweighs the positive $\Delta H$.
>
> Human Evaluation: answer is Correct but Unsupported by the grounding context as its missing the $\Delta G = \Delta H - T\Delta S$ equation.

- **Computer science**: 18/20 question/answer pairs correct, and 18/20 with correct and valid grounding in the source contexts. Both incorrect question/answer pairs were judged as failures because the question could not stand independent of the grounding context, as the question referenced information in the grounding context. These two grounding failures caused the questions to be labeled incorrect.

- **Math**: 18/20 question/answer pairs correct, and 18/20 with correct and valid grounding in the source contexts. One of the two incorrect questions was the result of the question not restating required information from the context. The other incorrect question was more interesting, as the question-writing model wrote an answer that is open-ended and non-unique:

> **Question 7:** Construct a rational equation that, after multiplying by its least common denominator, yields an algebraic solution which is exactly an excluded value, and explain why this equation has no real solution despite the apparent solution.
>
> **Answer:** $(x-1)/(x-1) = 2$; the only solution $x = 1$ is excluded because it zeroes the denominator, so there is no real solution.
>
> **Human Evaluation:** answer is Incorrect and Supported by the grounding context, because the provided answer is one possible answer of many.

Both `math` subset grounding failures were a result of the question writing model adding additional correct information from its parameter knowledge to flesh out the question.

- **Psychology**: 18/20 question/answer pairs correct, and 19/20 with correct and valid grounding in the source contexts.

Out of the 120 human evaluated question/answer pairs, 111 were correct. This gives the generative benchmarking an approximate 7.5% noise floor. 104 of 120 questions had fully supported grounding in the document. The vast majority of the grounding failures fell into two categories: 1) the question writing

model adding relevant correct information that was missing from the document, or 2) the generative question referencing the document and not being fully independent.

## B.1 Detailed Human Evaluation Results: MMLU-pro Physics

In order to provide detailed insight into the questions and answers involved in the human evaluation, we reproduce the set of 20 manually evaluated physics questions here, along with the human judgment about question/answer quality.

---

**Question 1:** Explain why operating a hair dryer (or similar bathroom appliance) while a bathtub or basin is filled with water creates a significant shock hazard, focusing on how the water's conductivity influences the formation of a ground-fault current path that can pass through a person.
**Answer:** Because water provides a low-resistance path that can complete a fault circuit, allowing current from the appliance to flow through a person in contact with the water.
**Human Evaluation:** Correct and Supported

**Question 2:** How does the difference in attachment distance of the back muscles compared with the quadriceps alter their mechanical advantage during a lift, and why does using the quadriceps to raise the load markedly reduce the compressive force on the vertebrae as illustrated in the example?
**Answer:** Because back muscles have a short lever arm (low mechanical advantage) while quadriceps have a longer lever arm (high mechanical advantage), using the legs shifts the load to stronger joints and markedly lowers the force on the spine.
**Human Evaluation:** answer is Correct and Supported by the grounding context.

**Question 3:** In a circuit where a switch is mistakenly wired in parallel with a resistive load, why does the load stay illuminated when the switch is open but go dark when the switch is closed, using Kirchhoff's current law and the notion of a short circuit to justify the behavior?
**Answer:** With the switch open, KCL forces all current through the load giving it full voltage; closing the switch creates a short-circuit branch that shunts the current, leaving near-zero voltage across the load so it extinguishes.
**Human Evaluation:** answer is Correct and Supported by the grounding context.

**Question 4:** For the 50.0 L high-pressure gas cylinder described (initial pressure $P_i$, initial temperature $T_i$), determine (a) the pressure after isochoric cooling to dry-ice temperature $T_{dry}$, (b) the pressure if 90% of the gas escapes, (c) the temperature to which the cylinder must be cooled to lower the pressure to 1 atm without any gas loss, and (d) whether such cooling is a practical safety strategy.
**Answer:** After cooling $P = P_i \cdot (T_{dry}/T_i)$; after 90% loss $P = 0.1 P_i \cdot (T_{dry}/T_i)$; to reach 1 atm, $T = (1 \text{ atm}/P_i) \cdot T_i$, a temperature far too low for practical use, making cooling impractical.
**Human Evaluation:** answer is Correct but Unsupported by the grounding context because its missing the governing relationships (ideal gas law / proportionalities).

**Question 5:** Considering the decay processes of neutral and charged pions outlined, which component of an extensive air shower that spreads over several square kilometers retains the greatest share of the primary cosmic-ray energy, and why?
**Answer:** The electromagnetic component, because $\pi^0$ decays into photons that dominate the cascade.
**Human Evaluation:** answer is Correct but Unsupported by the grounding context because its missing the cosmic ray shower component energy shares.

**Question 6:** Starting from the continuity of pressure and particle velocity at an interface between two media, derive the intensity reflection coefficient for an ultrasound wave and explain the physical significance of the numerator, denominator, and the squaring operation in the resulting expression.
**Answer:** $R = [(Z_2 - Z_1)/(Z_2 + Z_1)]^2$; $Z_1$ and $Z_2$ are the media's acoustic impedances, $(Z_2 - Z_1)$ reflects mismatch, $(Z_2 + Z_1)$ represents total impedance at the interface, and squaring yields the intensity ratio.
**Human Evaluation:** answer is Correct and Supported by the grounding context.

**Question 7:** In a mixed series-parallel network where a set of resistors is connected in parallel and that branch is in series with other elements, which tolerance extreme for the parallel resistors yields the highest total power dissipation, and why does this represent the worst-case overload condition?
**Answer:** When the parallel resistors are at their minimum tolerance (lowest resistance), because this minimizes equivalent resistance and maximizes total current and power.
**Human Evaluation:** answer is Correct and Supported by the grounding context.

**Question 8:** During the inelastic collision of a moving disk with a stick pivoted at one end, why does the system's angular momentum about the pivot remain conserved despite external forces, whereas its total linear momentum in the laboratory frame does not?
**Answer:** Because the external force passes through the pivot, giving zero torque (conserving angular momentum), but it supplies a net external impulse that changes linear momentum.
**Human Evaluation:** answer is Correct and Supported by the grounding context.

**Question 9:** Why can a pion exchanged between a proton and a neutron transmit the strong nuclear force without being directly observable, and how does its fleeting existence satisfy $\Delta E \cdot \Delta t \geq \hbar/2$ while leaving the total energy–momentum of the nucleon system unchanged?
**Answer:** Because the pion exists only for a time $\Delta t \leq \hbar/(2\Delta E)$ (with $\Delta E = mc^2$), allowing a temporary energy borrowing that is repaid when the pion is reabsorbed, so the nucleon system's total energy–momentum stays conserved, rendering the pion virtual and unobservable.
**Human Evaluation:** answer is Correct and Supported by the grounding context.

**Question 10:** When a cyclist applies a constant torque to the drivetrain, how does decreasing the wheels' moment of inertia affect the bicycle's linear acceleration, and what is the underlying reason?
**Answer:** Linear acceleration increases because a lower moment of inertia gives a higher angular acceleration for the same torque,

reducing the rotational energy share and directing more power into translational motion.
**Human Evaluation:** answer is Correct and Supported by the grounding context.

**Question 11:** When bending forward to touch the toes while keeping the heels and back against a wall, what torque must the hip and knee muscles generate about the heels to keep the system in static equilibrium, and how does selecting the wall contact point as the pivot affect this torque calculation?
**Answer:** A torque equal to the body's weight multiplied by its horizontal distance from the heels; choosing the wall contact as pivot removes this torque because the wall reaction provides an equal balancing moment.
**Human Evaluation:** answer is Incorrect but the question is Supported by the grounding context.

**Question 12:** Derive, using the subshell capacity formula $2(2l + 1)$, why the total number of electrons that can occupy all subshells of a shell with principal quantum number $n$ equals $2n^2$, and list the implicit assumptions required for this derivation.
**Answer:** Summing $2(2l + 1)$ for $l = 0$ to $n - 1$ yields $2n^2$; the derivation assumes the allowed $l$-range ($0 \leq l \leq n - 1$), full $m$-degeneracy $(2l + 1)$, spin degeneracy of 2, independent full filling of subshells, and neglects relativistic and electron-interaction effects.
**Human Evaluation:** answer is Correct and Supported by the grounding context.

**Question 13:** Using $\gamma = 1/\sqrt{1 - v^2/c^2}$, determine how the relativistic kinetic energy $K = (\gamma - 1)mc^2$ scales as $v$ approaches $c$, and state the limiting value of the classical kinetic energy $\frac{1}{2}mv^2$ in the same limit.
**Answer:** Relativistic $K$ diverges to infinity, while classical $K$ approaches $\frac{1}{2}mc^2$ (a finite value).
**Human Evaluation:** answer is Correct and Unsupported by the grounding context as its missing information about the classical limiting value.

**Question 14:** Which term describes chaotic systems that nevertheless develop organized, long-lasting structures, and what is the example of such a structure in Jupiter's atmosphere?
**Answer:** Stable chaos, exemplified by the Great Red Spot on Jupiter.
**Human Evaluation:** answer is Correct and Supported by the grounding context.

**Question 15:** Derive the decibel expression for sound-intensity level from the fact that human hearing responds logarithmically to intensity, and explain why the formula uses a factor of 10 for intensity ratios but a factor of 20 when expressed in terms of pressure ratios.
**Answer:** $\beta = 10 \log_{10}(I/I_0) = 20 \log_{10}(p/p_0)$ (the factor 20 arises because intensity $\propto$ pressure$^2$).
**Human Evaluation:** answer is Correct and Supported by the grounding context.

**Question 16:** Which analytical method, involving the decomposition of the initial velocity into horizontal and vertical components, is used to determine a projectile's range, and what explicit assumption about air resistance does the context provide for this calculation?
**Answer:** The analytical method that decomposes the initial velocity into horizontal and vertical components, with air resistance assumed zero.
**Human Evaluation:** answer is Correct and Supported by the grounding context but the question is degenerate and directly references the context.

**Question 17:** How can one determine the fraction of a human body that remains submerged when floating in freshwater versus seawater based solely on the body's overall density, and why does an increase in low-density fat relative to higher-density muscle cause that fraction to decrease (i.e., make the person float higher)?
**Answer:** The submerged fraction equals the body's density divided by the fluid's density ($\rho_{\text{body}}/\rho_{\text{water}}$ in fresh water and $\rho_{\text{body}}/\rho_{\text{seawater}}$ in salt water); increasing low-density fat lowers $\rho_{\text{body}}$, thus decreasing that fraction and raising the body higher in the water.
**Human Evaluation:** answer is Correct but Unsupported by the grounding context because its missing the density fraction formula.

**Question 18:** How does defining internal energy microscopically as the total kinetic and potential energy of a system's atoms and molecules justify the macroscopic first-law relation $\Delta U = Q - W$, and why does this make $\Delta U$ path-independent while heat and work remain path-dependent?
**Answer:** Because internal energy is a state function determined solely by the microscopic kinetic and potential energies of the particles, its change equals the net heat added minus the net work done ($\Delta U = Q - W$) for any process, making $\Delta U$ path-independent while $Q$ and $W$ are path-dependent.
**Human Evaluation:** answer is Correct and Supported by the grounding context.

**Question 19:** Considering Germany's 2020 renewable electricity share of 2.4% and the projected rise in global energy demand, what does achieving the 2030 target of 65% renewable power imply about its technical and economic feasibility?
**Answer:** It implies an extremely challenging technical and economic undertaking, requiring about a 27-fold increase in renewable share.
**Human Evaluation:** answer is Correct and Supported by the grounding context but the context table is missing some renewable information which makes the answer correct only w.r.t. the context. Additionally, the question is degenerate and uninteresting.

**Question 20:** A 10-cm pacemaker lead moving at 10 cm s$^{-1}$ perpendicular to an MRI scanner's magnetic field generates a Hall voltage of 20 mV. What magnetic field strength does this imply, and, accounting for the lead's inductance and the rapid switching of MRI gradient fields, is the induced current likely to disrupt pacemaker operation?
**Answer:** 2 T, and the induced current is negligible, so pacemaker interference is unlikely.
**Human Evaluation:** answer is Correct but Unsupported by the grounding context because its missing information about lead inductance and MRI switching.

# C    Impact of question and answer length

While longer more detailed questions (both MC and OE) produce a model over-performance (see Figure 6), the same trend does not appear in answer length. Figure 9 demonstrates this via scatterplot showing synthetic benchmark accuracy over-performance. Only the gpt-4.1-mini at the very right of the plot is significantly

biased above the $y = 0$ expected trend line. The over-performance trend shown for questions in Figure 6b is absent.

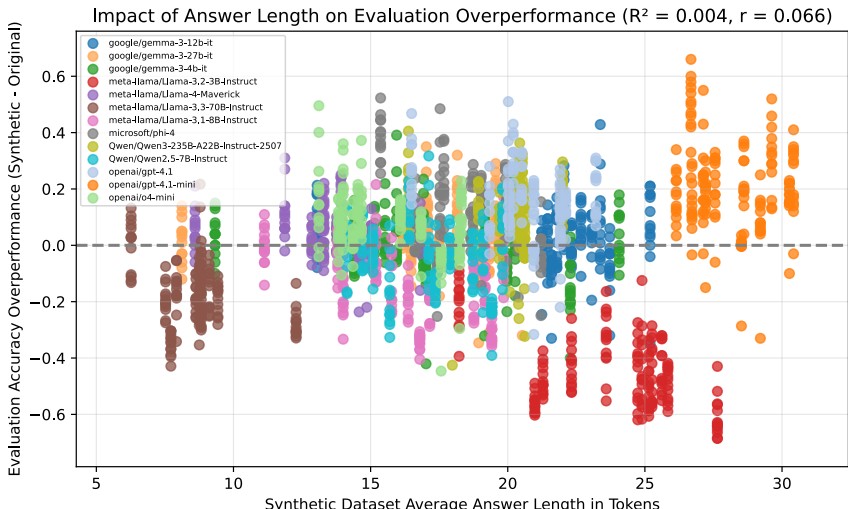

Figure 9: Impact of synthetic MCQ answer options token length on evaluation accuracy over-performance. LMs under evaluation do not over-perform when answer options are longer in the same way long questions cause over-performance.

The model gpt-4.1-mini demonstrates the largest over-performance for both question length and answer length as shown in Figure 6b. The following question answer pair was generated by gpt-4.1-mini (37 tokens) using a context from Squadv2, highlighting the difference in model verbosity in both question and answers:

```
1    "question": "How does the widespread accessibility and overuse of antibiotics, particularly in
         livestock, mechanistically contribute to the accelerated development of bacterial resistance,
         and what global health implications does this relationship entail?",
2
3    "choices": {
4      "A": "Overuse increases bacterial exposure to antibiotics, promoting selection for resistant
           strains, which leads to treatment failures and a global rise in antimicrobial-resistant
           infections.",
5      "B": "Limited antibiotic use in livestock reduces bacterial mutations, thereby decreasing
           resistance and minimizing global health risks.",
6      "C": "Antibiotic accessibility restricts bacterial gene exchange, preventing resistance
           development and protecting global health.",
7      "D": "Overuse of antibiotics in humans alone causes resistance, while use in livestock has no
           significant impact on bacterial evolution or public health."},
8
9    "answer": "A"
```

Contrast with this question answer pair generated by Llama-4-Maverick-17B-128E-Instruct using the same context and topic. Maverick produces a significantly shorter question (22 tokens) and answers:

```
1    "question": "What is the primary mechanism by which the widespread use of antibiotics in livestock
         contributes to the development of antibiotic-resistant bacteria in humans?",
2
3    "choices": {
4      "A": "By directly transferring resistant bacteria from animals to humans through the food chain."
           ,
5      "B": "By exerting selective pressure that favors the survival and proliferation of resistant
           bacterial strains.",
6      "C": "By altering the human gut microbiota, thereby reducing its ability to fight off infections
           .",
7      "D": "By inducing genetic mutations in human pathogens that confer resistance."},
8
9    "answer": "B"
```

# D   Synthetic benchmark question diversity

Most LMs produce similar diversity scores that are closely aligned with the human questions Some highly capable models (such as gpt-4.1) show comparatively worse question diversity while performing well in clarity,

correctness, and groundedness. Figure 10 shows the question diversity per generating model along with the original question diversity.

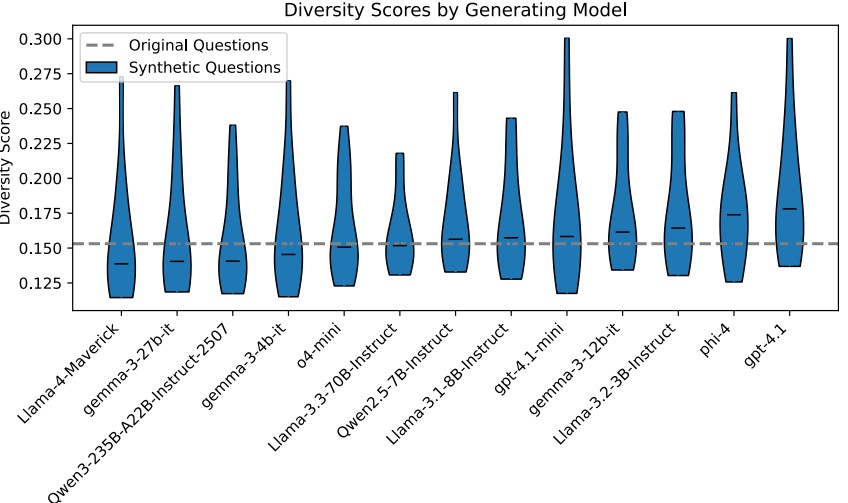

Figure 10: Synthetic question diversity (pairwise cosine distance) per generating model. Violin bars show the spread across the NLP datasets.

# E    Multiple-choice vs open-ended questions

The generating models which create overly easy questions by including too much detail (like phi-4 and gpt-4.1-mini) do so for both open ended and multiple choice questions. The long question problem becomes even more pronounced for open ended questions as shown in Figure 11.

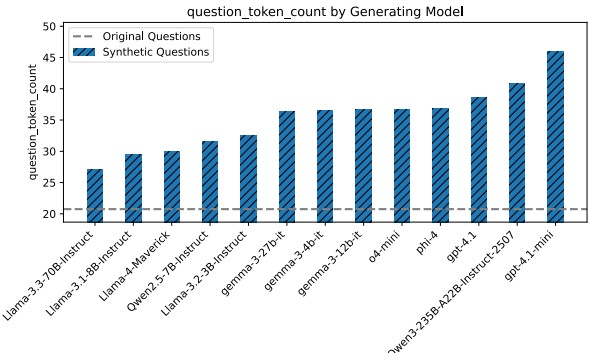

(a) MC question token count by generating model.

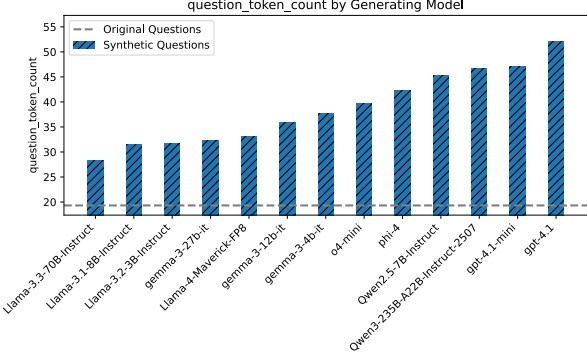

(b) OE question token count by generating model.

Figure 11: Most generating models have higher token counts than human, but some produce overly long questions. Both multiple-choice and opened-ended questions trend toward longer questions overall.

These overly long synthetic questions result in an over-performance by smaller models on the generated benchmark as shown in Figure 12. We hypothesize this happens because the additional detail in the question provides the LM under evaluation more information to pattern match against text seen in the training phase, thereby increasing the likelihood that the LM retrieves memorized patterns relevant to the question. This effect is seen in Figure 12a for multiple choice, and Figure 12b for open-ended. Both plots in Figure 12 demonstrate LM benchmark evaluation accuracy over-performance (synthetic − original accuracy).

While similar trends in question length and the over-performance that induces, the evaluation accuracy correlations and model rank order work equally well for open ended and multiple choice questions. Figure 13a

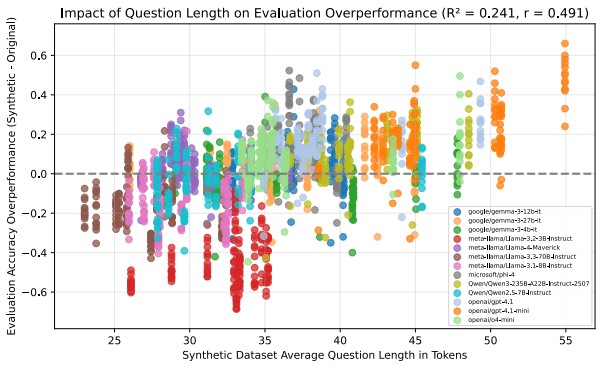

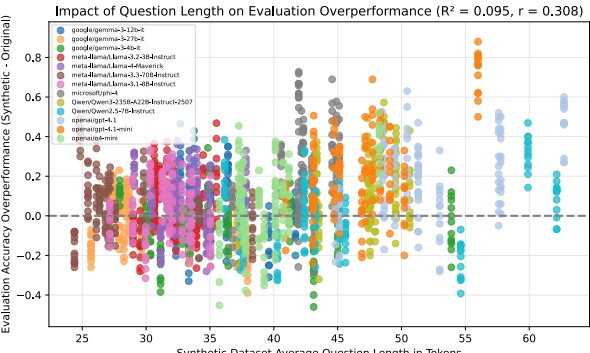

(a) Impact of multiple-choice question token count on evaluation accuracy over-performance, caused by overly easy questions.

(b) Impact of open-ended question token count on evaluation accuracy over-performance, caused by overly easy questions.

Figure 12: Longer synthetic open-ended questions have the same impact on evaluation accuracy over-performance as MCQ.

plots the correlation in evaluation accuracy between the human reformatted questions and the synthetic generated benchmark for multiple choice, and Figure 13b for open-ended.

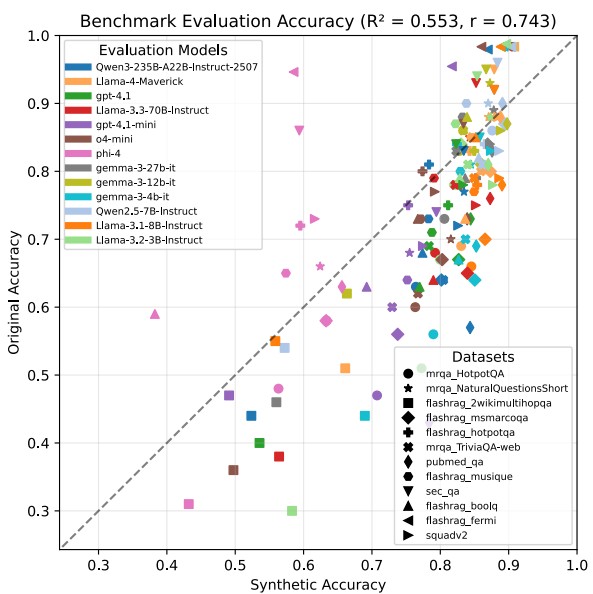

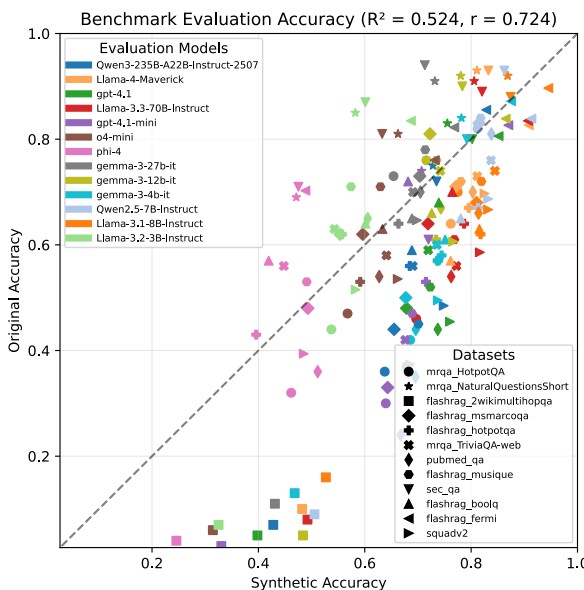

(a) Multiple Choice evaluation accuracy for the generating model ensemble.

(b) Open Ended evaluation accuracy for the generating model ensemble.

Figure 13: Multiple Choice vs Open Ended synthetic vs human benchmark evaluation performance exploring Pearson accuracy correlation. Both results demonstrate reasonable Pearson correlations between the benchmark evaluation accuracy.

# F  Evaluation benchmark accuracy and ranking correlation

Figure 13a shows the evaluation accuracy scatterplot comparing the reformatted human questions against the synthetic generated benchmark questions for the bolded ensemble models in Table 2. This ensemble produces a

Pearson accuracy correlation of 0.743 between the evaluation accuracies for synthetic and human benchmarks. Each plot marker is the average evaluation accuracy for a given model-dataset combination. While this model ensemble produces reasonable overall accuracy correlation, occasionally specific LMs generate undesirable benchmarks. Figure 14 showcases a failure where the synthetic benchmark generated by gpt-4.1-mini contains overly easy questions where all models perform near 100% evaluation accuracy (Pearson accuracy correlation 0.276.). This over-performance is likely linked to the question length problem.

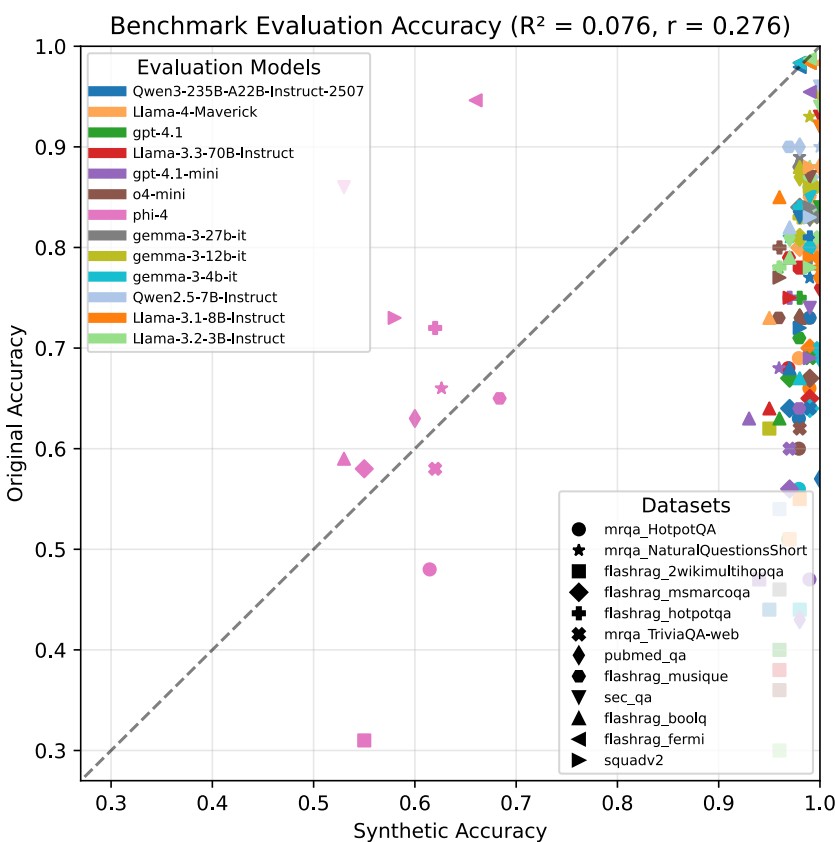

Figure 14: Synthetic over-performance for generating model gpt-4.1-mini indicating overly easy questions (Pearson accuracy correlation 0.276.)

Figure 15 plots the evaluation model performance across all datasets, comparing the synthetic and human accuracy Spearman ranking correlation of the evaluation model capability. This matches the use case of a user selecting between various models for a specific application relevant to the provided grounding documents. Figure 15a is the rank order barchart equivalent of Figure 13a and Figure 15b is the rank order barchart equivalent of the failure case in Figure 14.

Table 4 showcases the Spearman rank correlation between evaluation benchmarks written by humans and the synthetic data versions. The table includes data for all three of the human question reformatting models used to demonstrate the minimal impact that reformatting has on the overall high level results. The reformatting operation was evaluated against a subset of datasets including Squadv2, HotpotQA, TrivaQA-web, NaturalQuestionsShort, PubMedQA, and SecQA. Overall, the best outcome is to ensemble together a few questions generated by various reasonably capable LMs to avoid the known slight self-bias/self-preference issue. While we used Llama-4-Maverick as the text feature extractor for measures like groundedness and clarity, that model was unexpectedly dominant in these final results.

Table 5 contains the benchmark evaluation Pearson accuracy correlation between the reformatted human questions and the synthetic data generated questions. Similarly to the Spearman ranking, Llama-4-Maverick

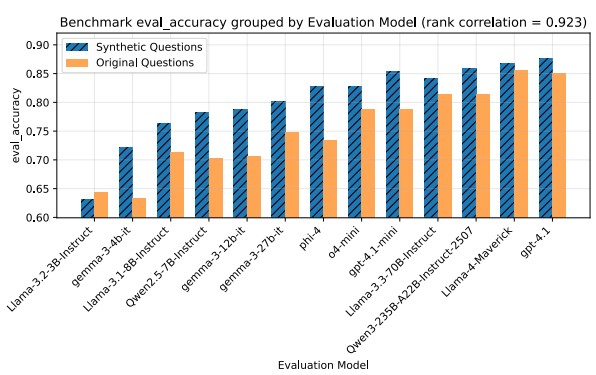

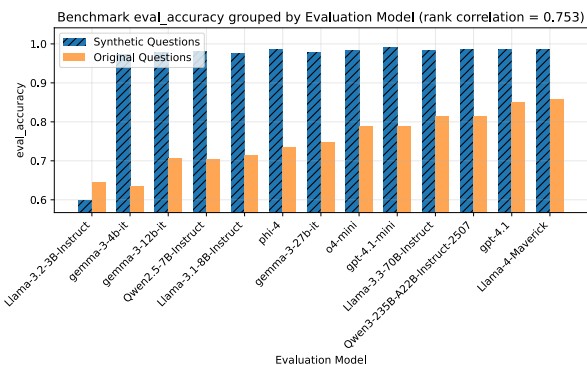

(a) Benchmark evaluation accuracy generated by Llama-4-Maverick producing a Spearman rank correlation 0.923.

(b) Benchmark evaluation accuracy generated by gpt-4.1-mini demonstrating over-estimation of LM capability with a Spearman rank correlation 0.742.

Figure 15: Comparison of benchmark rank ordering between synthetic and human evaluation accuracy.

Table 4: Spearman rank correlation between synthetic and human benchmarks for multiple choice (MC) and open-ended (OE) evaluations across generating models. Ensemble = bolded models.

| Reformat Model | Llama-4-Maverick | | gemma-3-27b-it | | o4-mini | |
|---|---|---|---|---|---|---|
| Generating Model | MC Rank | OE Rank | MC Rank | OE Rank | MC Rank | OE Rank |
| **Ensemble** | 0.9679 | 0.9209 | 0.9429 | 0.9253 | 0.9821 | 0.8813 |
| Llama-3.2-3B-Instruct | 0.6143 | 0.7143 | 0.6929 | 0.7143 | 0.6536 | 0.6791 |
| **Llama-3.3-70B-Instruct** | 0.9571 | 0.8813 | 0.9643 | 0.8681 | 0.9714 | 0.8242 |
| Llama-3.1-8B-Instruct | 0.7964 | 0.7275 | 0.7321 | 0.8110 | 0.7714 | 0.7275 |
| **Llama-4-Maverick** | **0.9679** | **0.9648** | 0.9536 | **0.9516** | **0.9821** | **0.9209** |
| gemma-3-12b-it | 0.8929 | 0.8198 | 0.8964 | 0.8418 | 0.9000 | 0.7890 |
| **gemma-3-27b-it** | 0.9571 | 0.9516 | **0.9857** | 0.9473 | 0.9714 | 0.9121 |
| gemma-3-4b-it | 0.6393 | 0.9209 | 0.6571 | 0.8989 | 0.6857 | 0.8637 |
| **gpt-4.1** | 0.9179 | 0.8549 | 0.9250 | 0.8725 | 0.9357 | 0.8110 |
| gpt-4.1-mini | 0.5643 | 0.8418 | 0.6286 | 0.8549 | 0.6071 | 0.7934 |
| **o4-mini** | 0.9250 | 0.7934 | 0.9036 | 0.8374 | 0.9429 | 0.7714 |
| phi-4 | 0.6964 | 0.1780 | 0.6250 | 0.3011 | 0.7607 | 0.1736 |
| Qwen2.5-7B-Instruct | 0.7321 | 0.8901 | 0.7143 | 0.8945 | 0.7786 | 0.8505 |

performed very well in these tests. However, unlike the ranking data, using a ensemble of models produces significantly worse Pearson accuracy correlation numbers under some circumstances. Likely due to the long and overly easy questions written by gpt-4.1 and o4-mini.

Table 5: Pearson accuracy correlation between synthetic and human benchmarks for multiple choice (MC) and open-ended (OE) evaluations across generating models. Ensemble = bolded models.

| Reformat Model | Llama-4-Maverick | | gemma-3-27b-it | | o4-mini | |
|---|---|---|---|---|---|---|
| Generating Model | MC Corr | OE Corr | MC Corr | OE Corr | MC Corr | OE Corr |
| Ensemble | 0.7452 | 0.7095 | 0.7248 | 0.5630 | 0.6569 | 0.4798 |
| Llama-3.2-3B-Instruct | 0.4857 | 0.4709 | 0.5116 | 0.6181 | 0.4448 | 0.5083 |
| **Llama-3.3-70B-Instruct** | 0.6615 | 0.6034 | 0.7284 | 0.7464 | 0.6390 | 0.5798 |
| Llama-3.1-8B-Instruct | 0.6608 | 0.6217 | 0.7231 | 0.6776 | 0.6472 | 0.5795 |
| **Llama-4-Maverick** | **0.7891** | **0.7233** | **0.8755** | 0.8217 | **0.8109** | **0.7833** |
| gemma-3-12b-it | 0.6879 | 0.4085 | 0.7748 | 0.5809 | 0.6986 | 0.4617 |
| **gemma-3-27b-it** | 0.7477 | 0.6962 | 0.8499 | **0.8292** | 0.7426 | 0.7046 |
| gemma-3-4b-it | 0.6575 | 0.5787 | 0.7117 | 0.6932 | 0.6524 | 0.6040 |
| **gpt-4.1** | 0.6188 | 0.5216 | 0.7218 | 0.6760 | 0.6528 | 0.5236 |
| gpt-4.1-mini | 0.4015 | 0.5780 | 0.4795 | 0.6508 | 0.4190 | 0.5271 |
| **o4-mini** | 0.6542 | 0.5700 | 0.7346 | 0.6430 | 0.6904 | 0.5503 |
| phi-4 | 0.1547 | 0.4965 | 0.2348 | 0.4951 | 0.2379 | 0.4043 |
| Qwen2.5-7B-Instruct | 0.5813 | 0.5905 | 0.6448 | 0.7199 | 0.5062 | 0.6019 |

## G   Question clarity and groundedness

We observed two failure modes with respect to grounding: Either the model hallucinates content absent from and not relevant to the context chunk (phi-4 commonly did this), or the generating model uses knowledge in its weights to expand upon the information presented in the context (o4-mini). For example, phi-4 generated the following question from the provided context:

```
1    "question": "How did the educational policies of the British Empire during the late 19th and early
        20th centuries shape the leadership qualities necessary for navigating modern challenges in the
        21st century?"
2
3    "context": "Victoria later described her childhood as \"rather melancholy\". Her mother was
        extremely protective, and Victoria was raised largely isolated from other children under the so
        -called \"Kensington System\", an elaborate set of rules and protocols devised by the Duchess
        and her ambitious and domineering comptroller, Sir John Conroy, who was rumoured to be the
        Duchess's lover. The system prevented the princess from meeting people whom her mother and
        Conroy deemed undesirable (including most of her father's family), and was designed to render
        her weak and dependent upon them. The Duchess avoided the court because she was scandalised by
        the presence of King William's bastard children, and perhaps prompted the emergence of
        Victorian morality by insisting that her daughter avoid any appearance of sexual impropriety.
        Victoria shared a bedroom with her mother every night, studied with private tutors to a regular
         timetable, and spent her play-hours with her dolls and her King Charles spaniel, Dash. Her
        lessons included French, German, Italian, and Latin, but she spoke only English at home."
```

## H   Case study on academic papers

Our approach was demonstrated on two academic papers: "Current Solutions and Future Trends for Robotic Prosthetic Hands" (labeled `annurev-control-071020-104336`) (Mendez et al., 2021), "Recent Advances in Large Language Model Benchmarks against Data Contamination: From Static to Dynamic Evaluation" (labeled `arXiv_2502_17521v1`) (Chen et al., 2025) and the recently released "America's AI Action Plan" (AiP). Figure 16 showcases LM benchmark evaluation performance on these two datasets; the MCQ results show most models performing well before a sharp dropoff with 4B and fewer parameter models and the Open-Ended results with a much smoother decline in performance. Multiple-choice questions can be easier to answer due to the presence of the answer options, even if the evaluation framework shuffles the option order. This effect is likely the cause of the MCQ evaluation accuracy to remaining higher for longer.

The following example questions were drawn from the MCQ synthetic benchmark constructed from `arXiv_2502_17521v1` (Chen et al., 2025).

```
1    {"question": "What is the primary distinction between the \"temporal cutoff\" approach and the \"
        rule-based generation\" approach in dynamic benchmarking?",
2
3      "choices": {
4        "A": "Temporal cutoff relies on newly released information, while rule-based generation creates
            novel evaluation data points using predefined rules.",
```

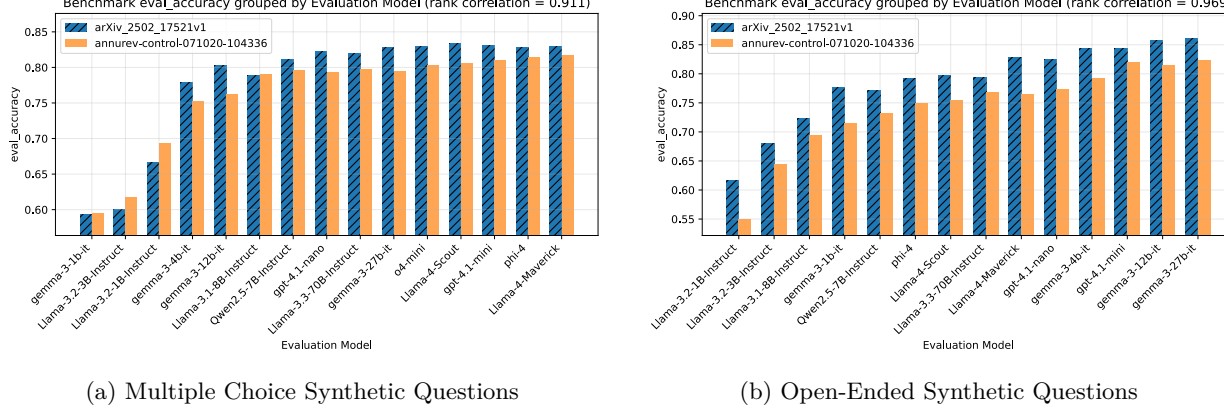

(a) Multiple Choice Synthetic Questions      (b) Open-Ended Synthetic Questions

Figure 16: Evaluation benchmark rank ordering comparing Multiple-Choice and Open-Ended questions performance across the arXiv_2502_17521v1 and "annurev-control" datasets. (Mendez et al., 2021; Chen et al., 2025).

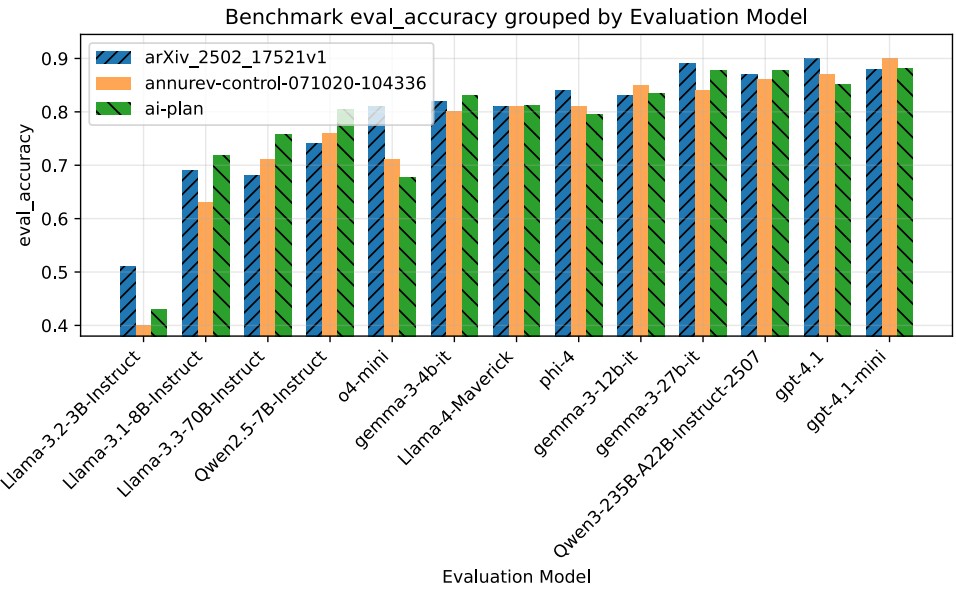

Figure 17: Evaluation benchmark rank ordering comparing Open-Ended questions performance across for three example documents. The arxiv paper and AI Action plan were released after most model training data cutoffs. (Mendez et al., 2021; Chen et al., 2025; AiP).

```
5        "B": "Temporal cutoff uses predefined rules, while rule-based generation relies on newly
             released information.",
6        "C": "Temporal cutoff is a hybrid approach combining multiple methodologies, while rule-based
             generation is a standalone method.",
7        "D": "Temporal cutoff generates data points using LLMs, while rule-based generation uses manual
             curation."},
8
9      "answer": "A"},
10
11    {"question": "What is a potential future research direction for improving the design and
           standardization of dynamic benchmarking methods for Large Language Models (LLMs), given the
           current limitations and proposed evaluation criteria?",
12
13      "choices": {
14       "A": "Developing more sophisticated data regeneration techniques to further minimize data
             contamination.",
15       "B": "Enhancing static benchmarking methods with more robust data encryption.",
16       "C": "Proposing new model architectures to reduce reliance on Internet-sourced training data.",
17       "D": "Focusing solely on post-hoc contamination detection methods."},
18
19      "answer": "A"}
```

The following example questions were drawn from the Open-Ended synthetic benchmark created for `arXiv_2502_17521v1`.

```
1    {"question": "How might the presence of contaminated benchmarks influence the trajectory of Large
           Language Model research and development, particularly in terms of model comparisons and
           deployment decisions?",
2
3      "answer": "By leading to misleading conclusions that favor contaminated models."},
4
5    {"question": "What are the implications of using LLMs to rewrite benchmark samples while preserving
            their original difficulty levels, as seen in methodologies like ITD, on the overall diversity
            and complexity of the resulting benchmark datasets?",
6
7      "answer": "The resulting datasets may maintain complexity but risk lacking diversity if not
             combined with other rewriting strategies."}
```

The following example questions were drawn from the Open Ended synthetic benchmark constructed from `US AI Action Plan` (AiP).

```
1    {"question": "What is the primary rationale behind requiring institutions receiving Federal funding
            to use nucleic acid synthesis tools with robust nucleic acid sequence screening and customer
            verification procedures?",
2      "answer": "To prevent the misuse of nucleic acid synthesis for harmful purposes."
3    },
4    {"question": "What would be the likely consequence for U.S. national security if the U.S. fails to
           impose comprehensive export controls on semiconductor manufacturing subsystems and fails to
           align its export control measures with those of its allies?",
5      "answer": "Adversaries could exploit U.S. semiconductor innovations, undermining U.S. national
             security."
6    },
7    {"question": "What would be a critical consideration in transforming NIST's Guardians of Forensic
           Evidence deepfake evaluation program into a widely adopted formal guideline for forensic
           analysis?",
8      "answer": "Ensuring adaptability to evolving deepfake technologies."
9    },
10   {"question": "How does the National AI Research Resource (NAIRR) pilot aim to enhance access to
           computing resources and AI models for the research community, and what are its potential
           implications for the development of a healthy financial market for compute?",
11     "answer": "By partnering with industry to provide access to world-class computing resources and
             building a sustainable operations capability."
12   },
13   {"question": "What are the potential benefits and challenges of interagency collaboration among
           departments like DOL, DOC, ED, and NSF in developing national skill frameworks for AI-related
           infrastructure occupations, and how might this collaboration impact the effectiveness of
           workforce development initiatives?",
14     "answer": "Interagency collaboration can facilitate a comprehensive approach to workforce
             development by aligning different agencies' expertise and resources, though it may be
             challenged by bureaucratic and coordination issues."
15   },
16   {"question": "What are the anticipated long-term implications of Executive Orders 14277 and 14278
           on the American workforce's ability to adapt to an AI-driven economy?",
17     "answer": "Enhanced adaptability and resilience through improved AI literacy and skills
             development."
18   }
```

# I   Evaluating LM self preference

When understanding potential sources of error for LLM-as-a-Judge how the information is presented to be evaluated is critical. The model evaluation requires the final model answer be on the last line starting with "ANSWER:" to make extraction easy.

With the question and model answer text extracted, it can be presented to the LLM-as-a-Judge for grading. The following prompt is used to evaluate answer correctness.

```
 1    {"You are comparing a submitted answer to an expert answer on a given question. Here is the data:
 2
 3        [BEGIN DATA]
 4        ************
 5        [Question]: What is a significant topological implication of a space being connected yet becoming
                totally disconnected upon removal of a single point, as exemplified by the Knaster-
                Kuratowski fan?
 6        ************
 7        [Expert]: It highlights the crucial role of a single point in maintaining the connectedness of
                the space.
 8        ************
 9        [Submission]: ANSWER: The removal of a single point can have a significant impact on the
                connectivity of a space, leading to unpredictable and drastic consequences for the space's
                topological properties.
10        ************
11        [END DATA]
12
13        Compare the factual content of the submitted answer with the expert answer. Ignore any
                differences in style, grammar, or punctuation.
14
15        Does the submission contain the content in the expert answer?
16
17
18        After assessing the submitted answer, reply with 'GRADE: $LETTER' (without quotes) where LETTER
                is one of CI.  Please choose ONE option for the grade: either \"C\" for correct answers, or
                \"I\" for incorrect answers.
19
20        For example, after reviewing a correct answer you might write 'GRADE: C' or after reviewing an
                incorrect answer you might write 'GRADE: I'.
21
22        First, write out in a step by step manner your reasoning about the criterion to be sure that your
                conclusion is correct. Avoid simply stating the correct answers at the outset. Then, end
                with your answer formatted as 'GRADE: $LETTER' (without quotes) where LETTER is one of CI."
23    }
```

This is then evaluated by the grader (with or without chain of thought) to produce a final output `"GRADE: CI"`.

```
 1    {"To assess whether the submitted answer contains the content in the expert answer, we need to
            analyze both answers and compare their factual content.
 2
 3        1. **Understanding the Expert Answer**: The expert answer states that a significant topological
                implication of a space being connected yet becoming totally disconnected upon removal of a
                single point is that it \"highlights the crucial role of a single point in maintaining the
                connectedness of the space.\" This indicates that the expert answer is emphasizing the
                importance of a particular point in keeping the space connected.
 4
 5        2. **Understanding the Submitted Answer**: The submitted answer suggests that \"the removal of a
                single point can have a significant impact on the connectivity of a space, leading to
                unpredictable and drastic consequences for the space's topological properties.\" This implies
                that removing a single point can greatly affect the space's connectivity.
 6
 7        3. **Comparing the Factual Content**:
 8        - The expert answer focuses on the role of a single point in maintaining connectedness.
 9        - The submitted answer discusses the impact of removing a single point on the space's
                connectivity and topological properties.
10
11        4. **Analyzing the Content**: Both answers touch on the idea that a single point is crucial for
                the connectedness of a space. The expert answer directly states this in the context of
                maintaining connectedness. The submitted answer conveys a similar idea by mentioning the
                significant impact of removing a single point on connectivity, although it generalizes this
                to topological properties as well.
12
13        5. **Determining Equivalence in Content**: The core idea in both answers is that a single point
                plays a critical role in the connectedness of a space. The submitted answer captures the
                essence of the expert answer by acknowledging the significant impact of removing a point on
                the space's connectivity.
14
15        6. **Conclusion**: The submitted answer conveys the same factual content as the expert answer
                regarding the importance of a single point for the connectedness of a space. Although the
```

```
        submitted answer is more general and does not limit itself to connectedness, it does contain
        the core idea presented in the expert answer.
16
17    GRADE: C"
18  }
```

One question that can be asked is do LMs under evaluation perform better on questions generated by the same model. In fact, the evaluation LMs do not over-perform on the questions they themselves generated. Figure 18 showcases the average generative evaluation accuracy across all datasets for pairs of question generation and evaluation LMs for open-ended questions graded by Llama-4-Maverick. The model-to-model difference in generation quality drives performance differences. In fact, when the generating model and evaluation model are the same is rarely the highest performing version of the questions.

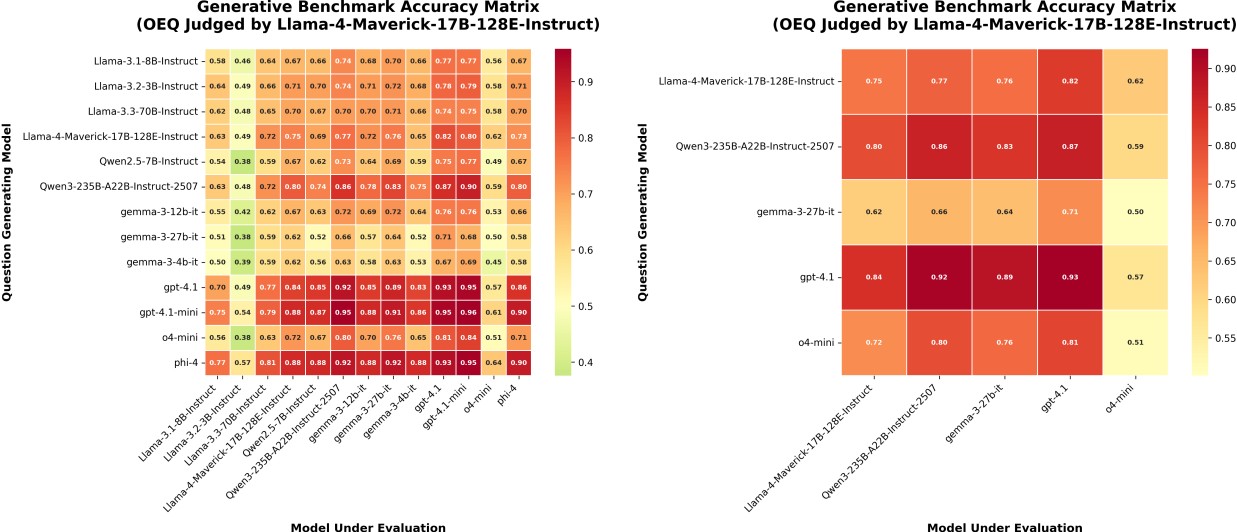

(a) Model average benchmark evaluation accuracy across all datasets. Open-ended questions graded by Llama-4-Maverick.

(b) Model average benchmark evaluation accuracy across all datasets for a subset of high accuracy LMs. Open-ended questions graded by Llama-4-Maverick.

Figure 18: The model-to-model difference in generation quality drives performance differences, and no LM self-preference (where models over-perform on the questions they generate) is observed.

Figure 19 showcases the average generative evaluation accuracy across all datasets for pairs of question generation and evaluation LMs for open-ended questions graded by gemma-3-27b-it.

Finally, there is the concern that individual bias in LM-as-a-Judge can cause problems with evaluation measurements. Figure 20 presents a heatmap outlining the evaluation accuracy averaged across all datasets for various combinations of open ended question grading model and the model under evaluation. The most impactful element of LM-as-a-Judge on evaluation accuracy is the strength or quality of the grader model used. To reduce grading noise one could ensemble multiple strong LM-as-a-Judge outputs together under a voting scheme.

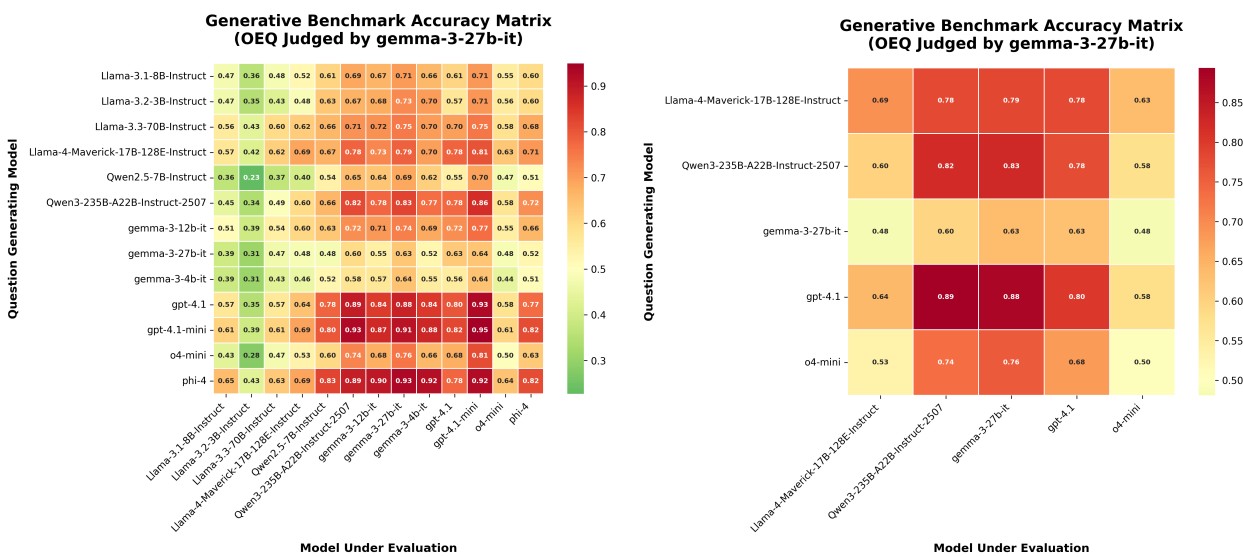

(a) Model average benchmark evaluation accuracy across all datasets. Open-ended questions graded by Llama-4-Maverick.

(b) Model average benchmark evaluation accuracy across all datasets for a subset of high accuracy LMs. Open-ended questions graded by Llama-4-Maverick.

Figure 19: The model-to-model difference in generation quality drives performance differences, and no LM self-preference (where models over-perform on the questions they generate) is observed.

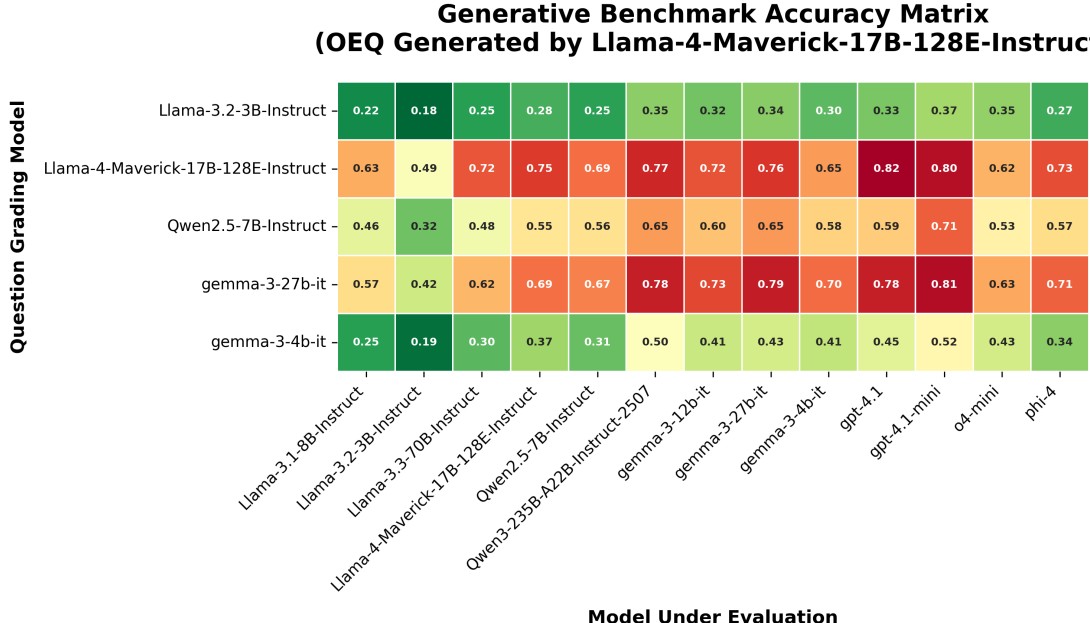

Figure 20: Heatmap showing the evaluation accuracy averaged across all datasets for various combinations of open ended question grading model and the model under evaluation. There is no strong self-preference for grading the models own output.

# J  Generative Benchmarking and Metadata Extraction Prompts

## J.1  Question Reformatting

When reformatting the varying quality of the extractive QA datasets the following prompt was used to clean up question references and create additional multiple choice answer options as outlined in Section 4.1.

```
1    # Your Role
2    You are an expert educational content creator specializing in crafting highly detailed evaluations
         to determine competency of topic domain experts based on the provided textual information. Your
          goal is to produce meaningful, highly challenging question-answer pairs that encourage
         reflection, insight, and nuanced understanding, tailored specifically according to provided
         instructions.
3
4    # Input Structure
5    Your input consists of:
6
7    <question>
8    [A question to be answered.]
9    </question>
10
11   <answer>
12   [The correct answer to the question.]
13   </answer>
14
15   <context>
16   [The text segment containing information relevant to the question.]
17   </context>
18
19   # Primary Objective
20   Your goal is to reformat, rephrase, and rewrite the question and answer pair according to the
         provided instructions. The rewritten question should be semantically equivalent and identical
         to the original question, rewritten for clarity. The rewritten answer should be semantically
         equivalent and identical to the original answer. Do not add any additional details to the
         question or answer that were not present in the original question or answer.
21
22   # Analysis Phase
23   Conduct careful analysis within '<document_analysis>' tags, following these steps:
24
25   1. Thoughtful Content Examination: Carefully analyze the given context, question, and answer;
         identifying central ideas, nuanced themes, and significant relationships within it.
26
27   2. Concept Exploration: Consider implicit assumptions, subtle details, underlying theories, and
         potential applications of the provided information.
28
29   3. Strategic Complexity Calibration: Thoughtfully rate difficulty (1-10), ensuring easy questions
         are avoided.
30
31   4. Intentional Question Planning: Plan how the question can invite deeper understanding, meaningful
          reflection, or critical engagement, ensuring the question is purposeful.
32
33   ## Documentation in Analysis
34
35   - Clearly document the rationale in the '<document_analysis>' tags when identifying irrelevant or
         bogus content, explaining your reasons for exclusion or inclusion decisions.
36   - Briefly justify any decision NOT to generate questions due to irrelevance or poor quality content
         .
37
38   # Question Generation Guidelines
39   ## Encouraged Question Characteristics:
40
41   - Thoughtful Engagement: Prioritize creating questions that inspire deeper thought and nuanced
         consideration.
42   - High Complexity: Develop questions that challenge the domain expert, following the provided
         additional instructions.
43   - Deep Understanding and Insight: Ensure that the question and answers require a deep understanding
          of the content by a professional domain expert.
44   - Self-contained Clarity: Questions and answers should contain sufficient context, clearly
         understandable independently of external references.
45   - Educational Impact: Ensure clear pedagogical value, reflecting meaningful objectives and genuine
         content comprehension.
46   - Conversational Tone: Formulate engaging, natural, and realistic questions appropriate to the
         instructional guidelines.
47
48   # Output Structure
49   Present your final output strictly adhering the '<output_format>' tags.
50   <output_format>
51   Question: [ Question Text ]
52   A: [ Answer Option A ]
```

```
53    B: [ Answer Option B ]
54    C: [ Answer Option C ]
55    D: [ Answer Option D ]
56    Explanation: [Brief explanation of why the answer is correct]
57    Correct Answer: [Letter of correct answer (one of A, B, C, or D)]
58    </output_format>
59
60    Begin by thoughtfully analyzing the provided context within '<document_analysis>' tags. Then
          present the resulting formatted question answer pair clearly within '<output_format>' tags.
61
62    # Important Notes
63    - Strive to generate questions that inspire genuine curiosity, reflection, and thoughtful
          engagement.
64    - Maintain clear, direct, and accurate citations/explanations drawn verbatim from the provided
          context.
65    - Ensure complexity and depth reflect thoughtful moderation as guided by the additional
          instructions.
66    - Each "thought_process" should reflect careful consideration and reasoning behind your question
          generation.
67    - When generating questions, NEVER include phrases like 'as per the text,' 'according to the
          document,' or any similar explicit references. Questions should inherently integrate content
          naturally and stand independently without explicit references to the source material. Make sure
          that the question is answerable by a domain expert **without the context paragraph**.
68    - Do not include answer information in the question.
69    - Ensure rigorous adherence to output formatting and generate a single '<output_format>' tag block.
70    - Ensure that all four answer options are distinct.
71    - Verify that the answer options are unambiguous.
72    - Verify the correct answer is present.
73    - Verify that the question and answer are semantically equivalent to the original question and
          answer.
74
75    <context>{context}</context>
76    <question>{question}</question>
77    <answer>{answer}</answer>
```

## J.2   Topic Extraction

This is the topic extraction prompt to convert a single chunk of document grounding context into a series of question topics that benchmark evaluation questions should be created around per Figure 2.

```
1     # Your Role
2
3     You are an expert educational content creator specializing in crafting highly detailed evaluations
          to determine competency of topic domain experts based on the provided textual information. Your
          goal is to produce meaningful, highly challenging question-answer pairs that encourage
          reflection, insight, and nuanced understanding, tailored specifically according to provided
          instructions.
4
5     # Input Structure
6     Your input consists of:
7
8     <context>
9     [The text segment to analyze, understand, and generate questions about.]
10    </context>
11
12    # Primary Objective
13    Your goal is to generate a list of all relevant topics, facts, or information that should be
          included in an examination to evaluate how well a professional domain expert understands the '<
          context>'.
14    The topics should encourage a deep engagement with the content, critically reflect on implications,
          and clearly demonstrate understanding and competency.
15
16    # Analysis Phase
17    Conduct careful analysis within '<document_analysis>' tags, following these steps:
18
19    1. Thoughtful Content Examination: Carefully analyze the given context, identifying central ideas,
          nuanced themes, and significant relationships within it.
20
21    2. Concept Exploration: Consider implicit assumptions, subtle details, underlying theories, and
          potential applications of the provided information.
22
23    3. Strategic Complexity Calibration: Thoughtfully rate difficulty (1-10), ensuring easy questions
          are avoided.
24
25    4. Intentional Question Planning: Plan how the question topcis can invite deeper understanding,
          meaningful reflection, or critical engagement, ensuring the questions are purposeful.
26
27    # Additional Instructions for Handling Irrelevant or Bogus Information
28    ## Identification and Ignoring of Irrelevant Information:
```

```
29
30     - Irrelevant Elements: Explicitly disregard hyperlinks, advertisements, headers, footers,
           navigation menus, disclaimers, social media buttons, or any content clearly irrelevant or
           external to the core information of the text chunk.
31     - Bogus Information: Detect and exclude any information that appears nonsensical or disconnected
           from the primary subject matter.
32
33     ## Decision Criteria for Question Generation:
34
35     - Meaningful Content Requirement: Only generate question topics if the provided '<context>'
           contains meaningful, coherent, and educationally valuable content.
36     - Complete Irrelevance: If the entire '<context>' consists exclusively of irrelevant, promotional,
           web navigation, footer, header, or non-informational text, explicitly state this in your
           analysis and do NOT produce any question-answer pairs.
37
38     ## Documentation in Analysis:
39
40     - Clearly document the rationale in the '<document_analysis>' tags when identifying irrelevant or
           bogus content, explaining your reasons for exclusion or inclusion decisions.
41     - Briefly justify any decision NOT to generate questions due to irrelevance or poor quality content
           .
42
43     # Question Topic Generation Guidelines
44     ## Encouraged Question Topic Characteristics:
45
46     - Thoughtful Engagement: Prioritize creating questions that inspire deeper thought and nuanced
           consideration.
47     - High Complexity: Develop questions that challenge the domain expert, following the provided
           additional instructions.
48     - Deep Understanding and Insight: Ensure that the question topics require a deep understanding of
           the content by a professional domain expert.
49     - Self-contained Clarity: Questions and answers should contain sufficient context, clearly
           understandable independently of external references.
50     - Educational Impact: Ensure clear pedagogical value, reflecting meaningful objectives and genuine
           content comprehension.
51     - Full Coverage: Ensure the generated topics fully cover the set of possible evaluation topics for
           the context for which high quality questions can be generated.
52
53     # Output Structure
54     Present your final output strictly adhering the '<output_format>' tags.
55     <output_format>
56     Topic: [ Topic Text ]
57     Topic: [ Topic Text ]
58     ...
59     </output_format>
60
61     Begin by thoughtfully analyzing the provided context within '<document_analysis>' tags. Then
           present the resulting formatted question topics clearly within '<output_format>' tags.
62
63     # Important Notes
64     - Strive to generate question topics that inspire genuine curiosity, reflection, and thoughtful
           engagement.
65     - Maintain clear, direct, and accurate citations/explanations drawn verbatim from the provided
           context.
66     - Ensure complexity and depth reflect thoughtful moderation as guided by the additional
           instructions.
67     - Each "thought_process" should reflect careful consideration and reasoning behind your question
           topic generation.
68     - Ensure rigorous adherence to output formatting.
69     - Get as detailed as required to fully cover the information present in the context.
70     - The topic response should be a single sentence that fully describes the topic.
71
72     <context>{context}</context>
```

## J.3 Open Ended Question Generation

This is the open ended question generation prompt. It requires a context chunk and question topic about which to write the question.

```
1     # Your Role
2     You are an expert educational content creator specializing in crafting highly detailed evaluations
           to determine competency of topic domain experts based on the provided textual information. Your
            goal is to produce meaningful, highly challenging question-answer pairs that encourage
           reflection, insight, and nuanced understanding, tailored specifically according to provided
           instructions.
3
4     # Input Structure
5     Your input consists of:
6
```

```
 7    <context>
 8    [The text segment to analyze, understand, and generate questions about.]
 9    </context>
10
11    <question_topic>
12    [A topic around which the question should be generated.]
13    </question_topic>
14
15    # Primary Objective
16    Your goal is to generate a single highly insightful and probing question-answer pair from the
          single provided '<context>'. Aim for highly technical understanding to probe domain expert
          knowledge about the '<context>'. The question needs to encourage a deep engagement with the
          content, critically reflect on implications, and clearly demonstrate understanding and
          competency. Constructed questions must be highly challenging to even the smartest domain
          experts.
17
18    # Analysis Phase
19    Conduct careful analysis within '<document_analysis>' tags, following these steps:
20
21    1. Thoughtful Content Examination: Carefully analyze the given context, identifying central ideas,
          nuanced themes, and significant relationships within it.
22
23    2. Concept Exploration: Consider implicit assumptions, subtle details, underlying theories, and
          potential applications of the provided information.
24
25    3. Strategic Complexity Calibration: Thoughtfully rate difficulty (1-10), ensuring easy questions
          are avoided.
26
27    4. Intentional Question Planning: Plan how the question can invite deeper understanding, meaningful
           reflection, or critical engagement, ensuring the question is purposeful.
28
29    # Additional Instructions for Handling Irrelevant or Bogus Information
30    ## Identification and Ignoring of Irrelevant Information:
31
32    - Irrelevant Elements: Explicitly disregard hyperlinks, advertisements, headers, footers,
          navigation menus, disclaimers, social media buttons, or any content clearly irrelevant or
          external to the core information of the text chunk.
33    - Bogus Information: Detect and exclude any information that appears nonsensical or disconnected
          from the primary subject matter.
34
35    ## Decision Criteria for Question Generation:
36
37    - Meaningful Content Requirement: Only generate questions if the provided '<context>' contains
          meaningful, coherent, and educationally valuable content.
38    - Complete Irrelevance: If the entire '<context>' consists exclusively of irrelevant, promotional,
          web navigation, footer, header, or non-informational text, explicitly state this in your
          analysis and do NOT produce any question-answer pairs.
39
40    ## Documentation in Analysis:
41
42    - Clearly document the rationale in the '<document_analysis>' tags when identifying irrelevant or
          bogus content, explaining your reasons for exclusion or inclusion decisions.
43    - Briefly justify any decision NOT to generate questions due to irrelevance or poor quality content
          .
44
45
46    # Question Generation Guidelines
47    ## Encouraged Question Characteristics:
48
49    - Thoughtful Engagement: Prioritize creating questions that inspire deeper thought and nuanced
          consideration.
50    - High Complexity: Develop questions that challenge the domain expert, following the provided
          additional instructions.
51    - High Difficulty: Ensure that the question is very difficult to answer correctly, even for the
          smartest domain experts.
52    - Generalizable: The best questions require the synthesis of high level general understanding above
           and beyond the specific context.
53    - Deep Understanding and Insight: Ensure that the question and answers require a deep understanding
           of the content by a professional domain expert.
54    - Self-contained Clarity: Questions and answers should contain sufficient context, clearly
          understandable independently of external references.
55    - Educational Impact: Ensure clear pedagogical value, reflecting meaningful objectives and genuine
          content comprehension.
56    - Conversational Tone: Formulate engaging, natural, and realistic questions appropriate to the
          instructional guidelines.
57    - Short and Factual: Ensure that the question and answer are short and factual, and that the answer
           is a single phrase or sentence.
58
59    (You do not need to use every question type, only those naturally fitting the content and
          instructions.)
60
```

```
61   # Output Structure
62   Present your final output strictly adhering the '<output_format>' tags.
63   <output_format>
64   Question: [ Question Text ]
65   Explanation: [Brief explanation of why the answer is correct]
66   Correct Answer: [Short answer]
67   </output_format>
68
69   Begin by thoughtfully analyzing the provided context within '<document_analysis>' tags. Then
         present the resulting formatted question answer pair clearly within '<output_format>' tags.
70
71   # Important Notes
72   - Strive to generate a question that inspires genuine curiosity, reflection, and thoughtful
         engagement.
73   - Maintain clear, direct, and accurate citations/explanations drawn verbatim from the provided
         context.
74   - Each "thought_process" should reflect careful consideration and reasoning behind your question
         generation.
75   - When generating questions, NEVER include phrases like 'as per the text,' 'according to the
         document,' or any similar explicit references. Questions should inherently integrate content
         naturally and stand independently without explicit references to the source material. Make sure
         that the question is answerable by a domain expert **without the context paragraph**.
76   - NEVER include information in the question that could give away the answer.
77   - NEVER ask questions where the answer is obvious or apparent.
78   - Verify that the correct answer is in fact correct and the best version of that answer.
79   - Ensure rigorous adherence to output formatting and generate a single '<output_format>' tag block.
80
81   <context>{context}</context>
82   <question_topic>{topic}</question_topic>
```

## J.4 Answer Correctness and Validity Prompt

This prompt extracts the **answer correctness** and **answer explanation validity** properties outlined in Section 4.4.

```
1    # Your Role
2    You are an expert evaluator of educational content. Your goal is to produce meaningful, insightful
         knowledge about domain expert evaluations designed to determine competence and knowledge.
3
4    # Input Structure
5    Your input consists of:
6
7    <question>
8    [A question to be answered.]
9    </question>
10
11   <answer>
12   [The student's answer to the question.]
13   </answer>
14
15   <explanation>
16   [An explanation for why the answer is correct.]
17   </explanation>
18
19   <context>
20   [The text segment containing information relevant to the question.]
21   </context>
22
23   # Primary Objective
24   You will be evaluating and judging the whether the student's answer and their explanation of why
         their answer is correct makes sense and is logically valid.
25
26   Your goal is to judge whether the information presented in '<answer>' is in fact the correct answer
         to the '<question>' given the information in the '<context>' and whether the '<explanation>'
         for why the answer is correct is valid. The information in '<context>' and '<question>' can be
         assumed true, only the context of '<answer>' needs to be validated for correctness.
27
28   ## Metrics
29   1. Answer Correctness: Rate from 1 to 10 how correct the provided student answer is given the
         information in the '<question>' and '<context>'. A rating of 1 indicates the answer is
         incorrect. A rating of 10 indicates the answer is correct and complete.
30
31   2. Explanation Validity: Rate from 1 to 10 how valid the students '<explanation>' of their answer
         is. The '<explanation>' should explain their thinking and the information used to determine the
         correct answer given the context and question. Low ratings indicate the explanation is not
         valid, correct, or that there is some flaw in the thinking or logic of the student. High
         ratings indicate the explanation is valid, correct, and explains why the answer is what it is.
32
33   # Analysis Phase
```

```
34    Conduct careful analysis within '<document_analysis>' tags, following these steps:
35
36    1. Thoughtful Content Examination
37    - Carefully analyze the given context, identifying central ideas, nuanced themes, and significant
          relationships within it.
38
39    2. Concept Exploration
40    - Consider implicit assumptions, subtle details, underlying theories, and potential applications of
           the provided information.
41
42    # Output Structure
43    Present your final output strictly adhering the '<output_format>' tags.
44    <output_format>
45    Answer Correctness: [ Correctness Rating. Respond with a number in [1, 2, 3, 4, 5, 6, 7, 8, 9, 10]
           ]
46    Explanation Validity: [ Validity Rating. Respond with a number in [1, 2, 3, 4, 5, 6, 7, 8, 9, 10] ]
47    </output_format>
48
49    Begin by thoughtfully analyzing the provided context within '<document_analysis>' tags. Then
          present the resulting formatted question answer pair clearly within '<output_format>' tags.
50
51    # Important Notes
52    - Each "thought_process" should reflect careful consideration and reasoning behind your ratings.
53    - Ensure rigorous adherence to output formatting.
54    - If either the question or answer is missing, rate the answer correctness and explanation validity
           as 1.
55
56
57    <question>{question}</question>
58    <answer>{answer}</answer>
59    <explanation>{explanation}</explanation>
60    <context>{context}</context>
```

## J.5 Clarity, Difficulty, and Groundedness Prompt

This prompt extracts the **question clarity**, **question difficulty**, and **question groundedness** properties outlined in Section 4.4.

```
1     # Your Role
2     You are an expert evaluator of educational content. Your goal is to produce meaningful, insightful
          knowledge about domain expert evaluations designed to determine competence and knowledge.
3
4     # Input Structure
5     Your input consists of:
6
7     <question>
8     [A question to be answered.]
9     </question>
10
11    <answer>
12    [The correct answer to the question.]
13    </answer>
14
15    <context>
16    [The text segment containing information relevant to the question.]
17    </context>
18
19    # Primary Objective
20    You will be evaluating and judging the quality of test and evaluation questions across a variety of
           metrics. Your goal is to judge and evaluate the quality of various test and evaluation
           questions across a variety of metrics. The '<question>' and '<answer>' pair is grounded and
           drawn from the '<context>'.
21
22    ## Metrics
23
24    1. Clarity: Rate from 1 to 10 the clarity and comprehensibility (how understandable it is) of the
          provided '<question>'. A rating of 1 is unclear and cannot be understood or cannot be
          understood without the '<context>'. A rating of 10 is used for questions that are self
          contained, understandable, and coherent (even if the topic is complex and difficult). Questions
           that are missing information required to understand what is being asked rate a 1. "As of the
          2015 NFL season, how many Super Bowl titles had the Denver Broncos won?" is a 10. "What event
          in 1861 contributed to the temporary strength of republicanism in Britain during Queen Victoria
          's reign?" is a 10. "In which year was the country not a member of FIFA, as indicated in the
          table?" is a 1. "As of the census of 2000, how many families were residing in the city?" is a
          1.
25
26    2. Difficulty: Rate form 1 to 10 the difficulty of the '<question>'. A rating of 10 is reserved for
           questions which require a deep understanding of the question and what is being asked by a
          professional domain expert.
```

```
27
28    3. Groundedness: Rate form 1 to 10 how grounded the provided `<question>` is in the `<context>`. A
         rating of 10 requires the question and answer information can found within the `<context>`. A
         rating of 1 indicates the question and answer information is not present in the `<context>`.
         This metric is only concerned with information found in the `<context>`, not outside
         information. The more outside information (not contained in the `<context>`) that is required
         to answer the question, the lower the rating.
29
30    # Analysis Phase
31    Conduct careful analysis within `<document_analysis>` tags, following these steps:
32
33    1. Thoughtful Content Examination: Carefully analyze the given context, identifying central ideas,
         nuanced themes, and significant relationships within it.
34
35    2. Concept Exploration: Consider implicit assumptions, subtle details, underlying theories, and
         potential applications of the provided information.
36
37    # Output Structure
38    Present your final output strictly adhering the `<output_format>` tags.
39    <output_format>
40    Clarity: [ Clarity Rating (one of [1, 2, 3, 4, 5, 6, 7, 8, 9, 10] ) ]
41    Difficulty: [ Difficulty Rating (one of [1, 2, 3, 4, 5, 6, 7, 8, 9, 10] ) ]
42    Groundedness: [ Groundedness Rating (one of [1, 2, 3, 4, 5, 6, 7, 8, 9, 10] ) ]
43    </output_format>
44
45    Begin by thoughtfully analyzing the provided context within `<document_analysis>` tags. Then
         present the resulting formatted question answer pair clearly within `<output_format>` tags.
46
47    # Important Notes
48    - Each "thought_process" should reflect careful consideration and reasoning behind your ratings.
49    - Ensure rigorous adherence to output formatting.
50
51
52    <question>{question}</question>
53    <answer>{answer}</answer>
54    <context>{context}</context>
```

