# OpenReview forum: "Grounding Generative Evaluations of Language Models in Unsupervised Document Corpora"
_TMLR — Accepted by TMLR_

### Review · Reviewer_Y4Vd · 2025-10-19

**Summary Of Contributions:**

This paper proposes a novel methodology for automatically constructing fact-based synthetic evaluation benchmarks for language models (LMs), grounded solely in user-provided authoritative documents (e.g., textbooks, research papers). The core idea is to leverage LMs themselves to generate high-quality, domain-specific questions—both multiple-choice (MCQ) and open-ended (OE)—that probe factual knowledge strictly derived from the source material.

Key contributions:
- A fully automated pipeline for benchmark generation: document chunking, topic extraction, question/answer generation, and evaluation.
- Empirical validation across 12 standard NLP QA datasets, showing strong alignment between synthetic and human-written benchmarks.
- Systematic analysis of question quality (clarity, groundedness, correctness) and diversity across a wide range of generator LMs.
- Identification and analysis of major failure modes (e.g., long-question bias).

Key strengths:
- High practical utility: Enables rapid, low-cost creation of domain-specific benchmarks without expert annotation.
- Comprehensive experimentation: Section 5 presents a thorough analysis, including dataset-level breakdowns, generator comparisons, ablations on question formats (MCQ vs open-ended), and qualitative error studies and so on.

Key weaknesses:
- Limited scope of validation datasets: All 12 are extractive QA tasks; I assume users will also be interested in their domain-specific, detailed cases, e.g, reasoning-heavy, multi-hop, mathematical, or code-based benchmarks.
- Reliance on LM-as-judge remains undervalidated: More human calibration or statistical significance analysis would strengthen claims.

**Audience:**

Yes

**Audience Explanation:**

TMLR readers, especially those studying evaluation methodology, benchmark construction, and domain adaptation of LMs, would find this work directly relevant. The paper addresses a highly practical and current problem: how to generate and maintain domain-specific benchmarks automatically. Its empirical and methodological insights (e.g., comparing MCQ vs open-ended evaluation, analyzing LM-judge bias) would appeal both to academic researchers and applied ML practitioners building evaluation systems for evolving domains.

**Broader Impact Concerns:**

The paper does not include a Broader Impact Statement, though the work carries low direct ethical risk. Potential concerns include:
Misleading evaluations if low-quality synthetic benchmarks are deployed without validation and overreliance on automation could reduce human oversight in high-stakes domains (e.g., medical or legal QA).

**Claims And Evidence:**

Yes

**Claims Explanation:**

The central claim—that synthetic benchmarks correlate well with human benchmarks—is robustly supported by quantitative metrics (high rank correlation (≈0.9) and reasonable Pearson correlation (≈0.7) with human benchmarks) and visualizations (showing tight alignment in model performance trends)

**Requested Changes:**

I have a few questions which might be helpful and included in the next version:
- In the related work section, discuss the difference between this work and the contemporary works.
- In Fig 3, the worst model is Llama3.3-70b, even worse than Llama3.2-3b. Could you explain the potential reason?
- How do authors choose the candidate models?
- Why do authors use the same model for question generation and answering?
- For diversity score, authors claim gpt 4.1 shows comparatively worse question diversity, but in Fig 8, it has the highest diversity score. Could you explain the detail or misunderstanding?
- In multiple-choice vs open-ended questions, authors claim 'the models that are good at creating synthetic multiple choice questions are also good at creating open ended questions'. Could you provide quantitative analysis on the relationship between these two settings?
- In real life, would using RAG on the provided text/information source be a more common scenario?

---

### Review · Reviewer_a3Nn · 2025-10-28

**Summary Of Contributions:**

This paper proposes a method for generating closed-book knowledge-based language model benchmarks (both multiple choice and open-ended questions) based on documents. It involves breaking the documents into smaller contexts, asking language models to identify "topics" in each context, then using the LM to generate a question and answer for each context and topic. These questions and answers can be automatically evaluated relative to the original context, and can then be used as a closed-book knowledge evaluation for a new model without including the context. The authors find that performance on their synthetically-generated benchmarks correlates with performance on human-written questions (rewritten by a LM to be closed-book answerable), although some models are better at writing good benchmark questions than others.

Overall the idea of the approach is interesting and relevant, although the method itself seems fairly straightforward once you know the idea. The results are interesting, but currently there are many details missing about the specific prompts and metrics used which make it somewhat difficult to understand what was done.

**Additional Comments:**

I am somewhat confused about Section 5.8. Since this section involves building closed-book benchmarks based on academic papers that are beyond the training cutoff, is it expected that models would be able to perform well on this benchmark? Intuitively it seems like many of the facts required to answer these questions would not be known to any model, and some of the questions in appendix G look very difficult to answer without the context document.

**Audience:**

Yes

**Audience Explanation:**

Benchmark generation for language models is a relevant area for individuals in TMLR's audience. Although the approach is fairly straightforward, I still think it would be interesting to some members of the audience if the full details of the approach were clearly described.

**Broader Impact Concerns:**

No broader impact concerns.

**Claims And Evidence:**

No

**Claims Explanation:**

It seems like many of the claims in this paper depend strongly on the specific details of the prompts and other specific methodology choices used, and these choices are not clearly explained in the paper. For instance, the meaning of "topic", the instructions for how to generate questions, the methodology for grading open-ended responses, and the meaning of quality measurements like "groundedness" or "answer correctness" are not clearly explained in the paper.

The paper indicates that the authors plan to release code. This seems critical for making sure the results are meaningful and reproducible. It would also be very useful if all prompts used for all of the steps of the pipeline could be included in the appendix, to make it clear what each of the steps actually involves.

It is also somewhat unclear whether "Original Questions" throughout the paper refers to the original version of the questions or the rewritten version following Section 4.1. For instance, which question versions were used for the length measurements in Section 5.3?

Another concern: Since the "Original Questions" were originally intended as open-book questions and are rewritten as part of this process, it is not obvious to me that comparing performance to the rewritten questions is a good way to evaluate model performance. It would be useful to compare also to some established fact-based benchmark, to help identify whether the resulting benchmarks are high-quality relative to evaluations designed for the same purpose.

**Requested Changes:**

### Add details of how each step works and what each quality metric means, including prompts used
In order to understand what these results are showing, it would be important to have more information on what was actually done. I suggest including the prompts in the appendix. It would also be worthwhile to add a bit more information on some of the steps in the main paper (e.g. how are open-ended questions graded?)

### Compare to existing fact-based benchmark instead of rewritten extractive datasets (or more clearly state limitation in abstract / intro / conclusion)
As far as I can tell, the generated questions are only ever evaluated against rewritten versions of extractive question-answering datasets, and never compared to question-answering datasets that were designed for closed-book knowledge evaluation. To tell whether the generated synthetic benchmarks are indicative of model knowledge, it seems important to also compare the ranking on the synthetic benchmark to some human-designed benchmark that was explicitly designed for evaluating knowledge, since it could be the case that the rewritten extractive questions are not actually good indicators of model knowledge.

(In absence of this, I think it would be good to explicitly state from the beginning that correlation is being taken relative to some rewritten extractive question benchmarks rather than to a benchmark explicitly designed for knowledge evaluation.)

### Clarify which version of "Original Questions" is used
When various results in the paper refer to "Original Questions", it is not clear if they are referring to the actual original version of the questions, or the LM-rewritten versions. Please clarify this.

### Clarify correlation in Section 5.8
What is the Spearman rank correlation taken over in Section 5.8? The rest of the paper made me expect that this would be a correlation between the synthetic benchmarks and some other ground-truth questions, but it doesn't look like there are ground truth questions here. Is this correlation just a correlation between performance on some pair of the new papers? It would be good to clarify this.

### Other small suggestions
The paper currently mentions "the original questions" in section 3.1, but this is not explained until section 4. It would be good to add a brief note in section 3.1 explaining what "the original questions" refers to (e.g. "a set of human-written questions based on the same context documents, rewritten to allow them to be answered without the context.").

Most of the citations in this paper use the incorrect style; when embedded in text they should appear like "some text (Name YYYY)" rather than "some text Name (YYYY)". Consider using something like `\citep`.

Nit in sec 4.2: period instead of comma in "During benchmark creation. the same model"

---

> ### Author Response · Authors · 2025-11-06
> **Requested Quality Metric Changes**
>
> Thank you for the valuable feedback and the time you spent on the thorough review.
>
> # Requested Changes:
>
> ## Add details of how each step works and what each quality metric means, including prompts used
>
> - A link to the github repository will be included in the final camera ready version. We have modified the manuscript appendix to include all the prompts used. As two examples, here are the relevant subsections from the larger metadata extraction prompt about clarity and groundedness:
> 	```
> 	1. Clarity: Rate from 1 to 10 the clarity and comprehensibility (how understandable it is) of the provided <question>. A rating of 1 is unclear and cannot be understood or cannot be understood without the <context>. A rating of 10 is used for questions that are self contained, understandable, and coherent (even if the topic is complex and difficult). Questions that are missing information required to understand what is being asked rate a 1.
> 	2. Groundedness: Rate from 1 to 10 how grounded the provided <question> is in the <context>. A rating of 10 requires the question and answer information can found within the <context>. A rating of 1 indicates the question and answer information is not present in the <context>. This metric is only concerned with information found in the `<context>`, not outside information. The more outside information (not contained in the <context>) that is required to answer the question, the lower the rating.
> 	```
>
> - We have added the following to the section about benchmark evaluation to clarify that we are using the stock open-ended question grader from the Inspect-AI framework: "Open ended questions are graded using the default Inspect-AI framework support which uses a narrowly scoped LM-Judge to compare a candidate model output to the ground truth answer to determine correctness."
>
> ## Compare to existing fact-based benchmark instead of rewritten extractive datasets (or more clearly state limitation in abstract / intro / conclusion)
>
> - The datasets which exist in literature with tuples of (question, answer, context) are generally extractive QA tasks from NLP research. Ideally a dataset more akin to MMLU would have grounding context documents, to evaluate generative benchmarking correlation with human written questions, we would need data where a context already has QA pairs to compare the LM generated question against, which is difficult to find. Creating such a dataset would require significant additional scope beyond using existing datasets. A major improvement to the dataset would be to source, vet, and then leverage grounding documents which contain the same knowledge as the MMLU benchmark.
>
> - We agree that the extractive QA sourced questions are not representative of the frontier of model knowledge. The goal is not to replace gold standard high quality human curated evaluation benchmarks (like Humanities Last Exam or LabBench), but to provide initial capability evaluations grounded in documents to assess the expected model capability in specific domains.

---

> ### Author Response · Authors · 2025-11-06
> **Requested Clarification Changes**
>
> # Requested Changes:
>
> ## Clarify which version of "Original Questions" is used
>
> - When the paper refers to "original questions," it is referring to the lightly modified version which is answerable without the grounding context paragraph. We went to some effort to verify that the topic, content, and intent of the question were unmodified during that disambiguation pass; there is a reformat verification process (omitted from the paper for length) where after reformatting, semantic alignment between the original human questions and the lightly rewritten questions was evaluated both by an LM-Judge and embedding cosine similarity to validate question similarity. We added the following to the Methodology section to clarify our definition of original question. "This disambiguation rewrite was validated using both an LM-Judge and embedding cosine similarity to ensure the questions were not unduly altered. Throughout the rest of this work, original questions refers to these disambiguated version of the human authored questions."
>
> - To ensue the correctness of this claim, we ran the full generative benchmarking analysis pipeline on the unmodified human written questions. The accuracy correlation was mildly worse (rho=0.73, due to some questions being unanswerable without the context) which showed up as a lower overall accuracy for the benchmark using the unmodified questions. The unmodified questions were even shorter than the minimal disambiguated versions. Among the question metadata properties: question diversity did not change much when using the unmodified human questions, groundedness increased compared to the disambiguated human questions, and clarity dropped a bit. Overall, all results were noisier when using the human written questions without the rewrite, largely due to the data quality of the underlying datasets.
>
>
> ## Clarify correlation in Section 5.8
>
> - You are correct that within the academic papers, there are no original human curated questions to correlate the benchmarking results against. We can only compare relative model performance on the generative benchmarks. We added the following to clarify that: "These correlations reflect the trend of overall model capability, as both source documents are from post training cutoff, so models forced to puzzle out what the likely answers are. Unlike earlier results where the correlation was a comparison to human curated questions, these correlations only compare the LM benchmark performance between the two academic papers."
>
>
> ## Other small suggestions
>
> - Most of the citations in this paper use the incorrect style; when embedded in text they should appear like "some text (Name YYYY)" rather than "some text Name (YYYY)". Consider using something like \citep.
>
> 	Corrected, with thanks.
>
> - The paper currently mentions "the original questions" in section 3.1
>
> 	We modified the sentence in section 3.1 talking about the original questions to clarify this point.
>
> - Nit in sec 4.2: period instead of comma in "During benchmark creation. the same model"
>
> 	Fixed, with thanks for the detailed feedback.

---

> > ### Comment · Reviewer_a3Nn · 2025-11-14
> >
> > Thanks for your response. I cannot seem to see the changes you describe, did you upload a revision?
> >
> > I also wanted to clarify my previous comment *"Compare to existing fact-based benchmark instead of rewritten extractive datasets (or more clearly state limitation in abstract / intro / conclusion)"*. In your response, you state
> >
> > > The datasets which exist in literature with tuples of (question, answer, context) are generally extractive QA tasks from NLP research. Ideally a dataset more akin to MMLU would have grounding context documents, to evaluate generative benchmarking correlation with human written questions, we would need data where a context already has QA pairs to compare the LM generated question against, which is difficult to find. Creating such a dataset would require significant additional scope beyond using existing datasets. A major improvement to the dataset would be to source, vet, and then leverage grounding documents which contain the same knowledge as the MMLU benchmark.
> > >
> > > We agree that the extractive QA sourced questions are not representative of the frontier of model knowledge. The goal is not to replace gold standard high quality human curated evaluation benchmarks (like Humanities Last Exam or LabBench), but to provide initial capability evaluations grounded in documents to assess the expected model capability in specific domains.
> >
> > I agree that, if you wanted to directly compare LM-generated questions against human-generated questions for the same contexts, this would require contexts to already have QA pairs, and that this would be out of scope for this work. However, if you want to judge whether your evaluations actually "evaluate domain-specific knowledge" and "replicate existing human-curated evaluations" (as you state in the intro), the important thing is to show that your evaluations would be a good substitute for some human-curated domain-specific evaluation. This could be done without necessarily exactly matching contexts. For instance, you could pick a particular domain that has a domain-specific benchmark and also has extractive QA tasks, and see whether the performance on your synthetic benchmark correlates with performance on a benchmark that is actually designed to evaluate knowledge in that domain. If so, perhaps your benchmark could be a good substitute; if not, it indicates that your method may not be testing the same kinds of things that human-designed evaluations test for.
> >
> > Without having some result like this, I don't think there is enough evidence to say that your method can "replicate existing human-curated evaluations" or that it "corresponds well with human curated questions", because you are not comparing to the main thing you are trying to replicate, which (as I understand it) would be a human-written domain-specific benchmark testing domain knowledge.

---

> > > ### Author Response · Authors · 2025-11-21
> > > **Direct Comparison to Human Curated Benchmarks**
> > >
> > > Thank you for the clarification. To address this concern we added the following experimental result to the Results section.
> > >
> > > Additionally, a new revision with all of the changes has been uploaded.
> > >
> > > ```
> > > 5.9 Replicating a subset of MMLU-Pro
> > > The datasets used herein to validate the generative benchmarking approach were never designed to evaluate general LM knowledge. To alleviate this weakness and demonstrate the capability of this generative benchmark creation approach we replicated a subset of the MMLU-Pro benchmark (Wang et al., 2024b) using creative commons textbooks from OpenStax (Stafford & Flatley, 2018). Six of the categories from MMLU-Pro were replicated by using a related textbook as the grounding context {biology (Clark et al., 2018), chemistry (Flowers et al., 2019), computer science (Franchitti, 2024), math (Abramson, 2021), physics (Urone & Hinrichs, 2022), and psychology (Spielman et al., 2020)} using Llama4-Maverick to construct the generative benchmarks. The full suite of models in Table 2 were evaluated on both the MMLU-Pro subsets and the generative benchmarks for the same topics. This produced a Pearson accuracy correlation of 0.86 and a Spearman rank correlation of 0.91.
> > > ```

---

### Review · Reviewer_1kyN · 2025-10-31

**Summary Of Contributions:**

The paper proposes an automated framework for creating synthetic evaluation benchmarks for large language models using only grounding documents (e.g., textbooks, papers, or policy reports) instead of manually constructed question datasets. The authors argue that traditional human-built benchmarks—such as MMLU—are limited, static, and often contaminated by training data, making them less effective for evaluating modern LMs.

The proposed method begins by converting documents into structured text and dividing them into context chunks, from which the model extracts key topics. For each topic–context pair, a language model generates a fact-based question, a correct answer, and an explanation, ensuring that all content is grounded in the source material. These generated questions can take either multiple-choice or open-ended formats. The resulting synthetic benchmarks are then used to assess other LMs’ performance, with metrics such as accuracy and rank correlation compared to human-written benchmarks. This pipeline enables scalable, domain-specific, and automatically updated model evaluations with minimal human intervention.

The experiments in the paper evaluate the proposed synthetic benchmark generation method across multiple natural language processing QA datasets, including SQuADv2, HotpotQA, TriviaQA, PubMedQA, BoolQ, and SecQA, by comparing synthetic, document-grounded questions against human-authored benchmarks. Using various large language models such as Llama-3/4, Gemma-3, GPT-4.1, Phi-4, and Qwen3, the authors assess question quality (clarity, groundedness, correctness, and diversity) and benchmark validity through Pearson accuracy and Spearman rank correlations between synthetic and human evaluations. Results show strong alignment, with an ensemble of models achieving a 0.74 Pearson correlation and 0.91 Spearman rank correlation. The study also includes a case study on three academic and policy documents to test real-world applicability, revealing that Gemma-3 models excelled on open-ended questions.

**Audience:**

Yes

**Audience Explanation:**

- This paper will receive attention from TMLR readers who build or evaluate LLMs. This paper offers a way to create domain-specific benchmarks from any corpus.

- The pipeline automates chunking documents, extracting topics, and generating grounded multiple-choice questions would be useful for language model evaluation practitioners in writing tests for language models.

**Claims And Evidence:**

No

**Claims Explanation:**

- There is a lack of a formal description of the method. The method involves document chunking, topic extraction, and question generation. It would be better to describe a mathematical problem formulation and solution for this method.
- Moreover, these methods potentially involve many parameters. How these parameters are adjusted is unknown.
- The protocol of how human evaluates the questions is not clearly described.

**Requested Changes:**

- It would be better to provide a formal mathematical problem formulation and method description, using necessary notations and algorithmic boxes to describe the pipeline.
- It would be better to describe the human evaluation process.
- Are there any potential baselines? It would be better to include a comparison with naive construction methods or perform ablation studies.

---

> ### Author Response · Authors · 2025-11-06
> **Addressing Requested Changes**
>
> We thank the reviewer for the thoughtful feedback.
>
> # On Formal Problem Formulation and Method Description:
>
> We appreciate the reviewer's request for a more formal description of the method. We acknowledge the value formalization can impart when describing layered LM pipelines.
>
> First, we would like to clarify the difference between the set of operations performed to evaluate the LM generative benchmarking (things like the metadata property extraction of answer correctness, explanation validity, or question grounding scoring) and the minimal operations required to actually use the generative benchmarking in practice. The vast majority of the prompting and potential parameterization in this work is in validating non-degenerate question and answer generation, not in the actual generative benchmarking phase. Ensuring that generative benchmarking produces reasonable questions and correct associated answers is more complex than using the framework operationally. Using the evaluation framework only requires some form of document chunking, topic extraction, and then question generation per topic. To help clarify this point, we have added a new figure to the start of the Methods section which covers the algorithmic workflow.
>
> Within that set of operations only three parameters exist: the LM used to perform generation, the length of the document chunk, and the overlap between document chunks. Within the manuscript we evaluated the LM. The paper includes results of generative benchmarks from all models in Table 2 under a full factorial design of experiment configuration. However, we do recognize that the prompts themselves represent a significant amount of implicit parameterization, as do the data sources used to ground the benchmark generation; these will have outsized impacts on the resulting quality.
>
> For all of our results, topic extraction and question generation used a static set of prompts. In the current submission, these prompts are available in the supplementary code on github. To make this more transparent and accessible, we will add all prompts directly to the paper's appendix.
>
>
> # On the Human Evaluation:
>
> Due to the number of datasets, questions, and generating models, no systematic human evaluation of the generated questions is performed. While spot checking was performed at many stages, we relied on LM-judge operations to extract properties of the generated questions and answers. The main quality evaluation properties were question_clarity, question_groundedness, answer_correctness, and answer_explanation_validity. answer_explanation_validity is a secondary check to validate that question generating model's explanation is valid. Our primary motivation for using extractive question answering datasets for methodology validation is the tight coupling between the tuples of (context, human written question, human written answer).
>
> # On Baselines:
>
> The only similar baseline work of which we are aware is Auto-Bencher, which looked at using RL optimization to discover gaps in LM knowledge. While Auto-Bencher generates evaluations, it is not attempting to generate benchmarks from grounding documents, but instead explore and find model weakness given a topic and privileged information. Auto-Bencher is optimizing for question novelty, separability (between models under evaluation), and difficulty. It would likely score fairly poorly when evaluated based on correlation with human written questions, as it was not attempting to replicate a benchmark from grounding documents, but instead iteratively explore the knowledge space of an LM.
>
> While we do not have a chart of the results, we did some exploratory testing using a single pass of the generating model for each context and having it perform topic extraction and multiple question generation in a single operation. During that testing we noticed no difference in resulting quality, so we opted to keep the separate topic extraction and question generation phases in order to maintain the narrowest possible scope for each LM operation.

---

### Decision · Action_Editor_pzNU · 2026-01-04

**Recommendation:** Accept with minor revision

**Audience:**

Yes

**Audience Explanation:**

The topic is clearly an important one for evaluating LLMs.

**Claims And Evidence:**

Yes

**Claims Explanation:**

This paper aims to study an important task of automatically constructing fact-based synthetic evaluation benchmarks for LMs using only user-provided grounding documents (e.g., textbooks). The authors demonstrated that their generative benchmarking approach achieves high correlation with human-curated questions. The method consists of an intuitive pipeline of chunking, topic extraction, and question generation. While the method is simple and straightforward, the authors have demonstrated with comprehensive evaluation results across many evaluated models and datasets (multiple choice & open-ended).

In the initial reviews, the reviewers have pointed out several concerns, including the details about prompts and the lack of validation against human-curated knowledge benchmarks. In the authors' revision, they have addressed most of these concerns, and the reviewers generally found the paper to be much strengthened.

In the camera-ready version, the authors should address Reviewer a3Nn's remaining concerns (quoted from the final recommendation):

> The authors have added a very brief section 5.9 based on my feedback, which also compares their generated evaluations to a subset of MMLU-Pro (by computing two correlation numbers). However, this section is very brief with few details, and it does not seem like enough to justify the full weight of the central claims of the paper; just knowing a correlation score with no other context does not seem like very strong evidence about whether these evaluations would be high-enough quality to substitute for human-generated ones.

As the authors promised to release the code, the AE believed that these details would be available as well. The authors should ensure that these are included.

> Additionally, there is no human validation of the questions (as mentioned in the discussion with reviewer 1kyN) and no comparison of the generated questions to human-designed benchmark questions; all validation of the quality of the generated questions is based on LM judgements and correlation scores.

The AE believed that larger-scale human validation is not feasible for this work, but a small-scale human evaluation will indeed strengthen the results. Please include small-scale human validation results (e.g., on a few questions & models) in the revision.

---

> ### Author Response · Authors · 2026-01-25
> **Camera Ready**
>
> We have addressed both the AE concerns. We expanded the detail, methodology, and results for section 5.9 MMLU-pro replication using the generative benchmarking approach. This includes the addition of a scatterplot showing the per-model benchmark accuracy numbers comparing the original mmlu-pro questions against our generative reconstruction of the mmlu-pro subsets.
>
> The final camera-ready version also includes a discussion of the new human validation of 120 of the generative mmlu-pro replication questions (20 randomly selected questions from each of the 6 mmlu-pro subsets). The appendix was also expanded to include details about the human evaluation of the 120 question/answer pairs, including success and failure cases. Finally, all 20 manually graded generative mmlu-pro physics questions are included for the reader to get an understanding of the success and failure modes of generative benchmarking.
>
> Thank you for your time and effort reviewing.